

# Enhancing AppAuthentix recommender systems using advanced machine learning techniques to identify genuine and counterfeit android applications

Ramnath M.[1] and Yesubai Rubavathi C.[2]

[1] Department of Artificial Intelligence and Data Science, Ramco Institute of Technology, Rajapalayam, Tamil Nadu, India
[2] Department of Computer Science and Engineering, Saveetha Engineering College, Thandalam, Chennai, Tamil Nadu, India

## ABSTRACT

Smartphone app expansion needs strict security measures to avoid fraud and danger. This study overcomes this issue by identifying apps differently. This new solution uses convolutional neural network (CNN), natural language processing (NLP), and the strong AppAuthentix Recommender algorithm to secure app stores and boost customer confidence in the digital marketplace. Since the app ecosystem has grown, counterfeit and harmful applications have risen, threatening consumers and app merchants. These risks need advanced technology that can distinguish malware from legitimate apps. A complex prediction model using CNNs for image analysis, NLP for text feature extraction, and the novel AppAuthentix Recommender algorithm to properly identify legitimate and counterfeit mobile applications is the goal of this research. The whole strategy secures app stores and authenticates apps. The urgent need to safeguard app markets and users against unauthorized and hazardous programs sparked this study. Our cutting-edge solutions make mobile app consumers' digital lives safer and app marketplaces more trustworthy. CNN, NLP, and AppAuthentix Recommender yielded amazing results in this investigation. Mobile app authenticity may be estimated with 98.25% accuracy. This technology greatly improves app store security and enables mobile app verification. In conclusion, our work offers a novel way to app identification at a time of rapid mobile app development. CNN, NLP, and AppAuthentix Recommender have dramatically enhanced app store security. These new solutions may boost mobile app security and consumer confidence.

# INTRODUCTION

Within the swiftly growing realm of mobile apps, people frequently depend on reviews of apps to make well-informed choices regarding acquiring and utilizing an app. The evaluations, which contain valuable consumer feedback and opinions, provide a wealth of data that may be utilized to enhance app recommendations. At the same time, the

Corresponding author
Ramnath M., ramnath25@gmail.com

smartphone application ecosystem faces obstacles, particularly the widespread existence of counterfeit applications that imitate genuine ones to trick consumers. To address this complex issue, it is necessary to employ advanced technology solutions capable of efficiently analyzing large volumes of data to deliver precise app recommendations and distinguish between authentic and counterfeit applications.

Mobile app recommendations and distinguishing between authentic and counterfeit applications are crucial topics in computer science, specifically in the realms of natural language processing (NLP) and convolutional neural network (CNN). NLP techniques examine user evaluations and textual material associated with applications, extracting significant insights and attitudes that might inform recommendation algorithms. Additionally, CNN, which are commonly employed for image processing tasks, have been utilized to analyze app characteristics, interfaces, and other forms of non-textual data to differentiate between authentic and counterfeit apps. App recommendations are customized suggestions given to users to aid them in finding new or pertinent apps that correspond with their interests, usage habits, or preferences. These suggestions can be created using several methodologies, such as assessing a user's previous app interactions, employing cooperative filtering algorithms that take into account the preferences of comparable users, or utilizing content-based filtering that aligns the app's capabilities with user interests. The objective is to optimize the user experience by improving the efficiency and customization of app discovery based on individual preferences.

App reviews, however, are assessments created by users that depict firsthand encounters with an application. Reviews often comprise a textual analysis and an evaluation, frequently using a grading system that ranges from one to five stars. Reviews serve many functions: they furnish vital perspectives for prospective users to assess the caliber and appropriateness of an application, give developers input on user preferences and criticisms, and have the potential to impact the visibility and ranking of the application in app stores. User reviews play a vital role in the app ecosystem, allowing users to convey their contentment, report any problems encountered, and propose enhancements. They are of utmost importance to users as well as developers.

Studies have demonstrated that NLP may be efficiently utilized to assess application reviews for providing suggestions. Methods such as analysis of sentiment, modeling topics, and extraction of keywords are utilized to comprehend user sentiments and preferences. *Ning et al. (2014)* conducted research where they used topic modeling to classify app evaluations into several topics. This approach assisted in comprehending customer issues and preferences, thereby enhancing app recommendations. In a study conducted by *Guzman & Maalej (2014)*, sentiment analysis was utilized to evaluate app reviews and determine app ratings. The research revealed that user feelings had a substantial impact on app recommendations.

Counterfeit app identification using CNN entails examining the visual characteristics of the app, including icons, images, and layout patterns. In a study conducted by *El-Gayar et al. (2024)*, a system was proposed that utilizes CNN to distinguish between authentic and counterfeit application interfaces. This framework achieved notable levels of accuracy. The CNN model underwent training using a dataset consisting of app images. During this

process, it acquired the ability to identify tiny variations in design and layout that are frequently disregarded by ordinary users but serve as indicators of counterfeit apps.

The integration of NLP with CNN provides a more resilient solution for app recommendations and counterfeit detection. The capacity of NLP to comprehend and examine textual data complements the proficiency of CNN in handling visual information. Li et al. (2019) introduced a complete method that used NLP and CNN to assess both textual reviews and visual aspects of applications. This methodology enhanced the precision of app recommendations and the identification of counterfeit apps.

The significance and capabilities of mobile app recommendations, which are based on app reviews and utilize NLP and CNN to distinguish between genuine and counterfeit apps, are crucial for improving user experience, ensuring security, and building trust in the digital ecosystem. This research utilizes NLP to examine user reviews, exploring the underlying attitudes and opinions stated by users. It extracts significant insights from this analysis, which are then used to provide tailored app recommendations. This sophisticated strategy guarantees that consumers are guided towards applications that not only fulfill their requirements and preferences but also align with the overall favorable feedback from the user community. Simultaneously, employing CNN to analyze the visual components of applications is a powerful method for identifying counterfeit apps and protecting users from possible scams and security risks. This comprehensive study not only provides consumers with the ability to make well-informed decisions and a more secure application environment but also gives developers valuable input that they can utilize to improve their products. Moreover, it improves the operational effectiveness of app marketplaces by streamlining search and recommendation algorithms, hence enhancing the overall quality and dependability of the app ecosystem. The study effort has discovered two primary issues, which are outlined below.

Primarily, the excessive proliferation of applications across different marketplaces poses a difficulty for consumers to find programs that truly align with their tastes and requirements, therefore demanding a more advanced method for app suggestions. The capacity of NLP to examine and extract significant insights from information provided by users, such as reviews, offers a method to customize suggestions more precisely according to specific user preferences by comprehending the subtleties of user feedback.

Furthermore, the digital marketplace is inundated with counterfeit applications that imitate authentic ones, presenting substantial hazards to user security and privacy. These unauthorized applications have the potential to result in unauthorized access to sensitive information, monetary damages, and a decline in confidence in online networks. CNN's ability to scan visual material provides a viable approach for spotting counterfeit applications by detecting inconsistencies in their graphical features compared to authentic ones.

The motivation for this research challenge stems from the urgent requirement to improve user experience in exploring app marketplaces and to strengthen security measures against counterfeit software. The integration of NLP and CNN in this study signifies an advanced method that aims to enhance the customization and significance of app suggestions while also protecting the digital ecosystem against counterfeit apps. This

study is situated at the point where user experience improvement and cybersecurity cross in the app marketplace. It focuses on answering important requirements that have substantial consequences for users, developers, and platform operators.

The study conducted by *Yang et al. (2021)* centers around a method of extracting phrases from user requests in mobile application reviews. This emphasizes the need to investigate and utilize more detailed aspects of textual data, such as analyzing sub-phrases or semantic roles, to improve the accuracy of app suggestions and insights obtained from reviews. *Martens & Maalej (2019)* explore the issue of identifying and detecting unauthorized reviews in app stores, highlighting the need for incorporating effective methods of evaluating the legitimacy of reviews into app recommendation algorithms. Ensuring the dependability of the user-generated material that contributes to recommendation algorithms is essential for their efficacy. *Yang et al. (2021)* provides a framework for extracting user requests. However, there is a lack of integration of these findings into real-time, context-aware recommendation systems that can adjust to users' evolving demands and circumstances. *Martens & Maalej (2019)* discuss the topic of unauthorized reviews, which are intricately connected to the larger problem of counterfeit software. There is a need for study in creating techniques that can detect counterfeit applications inside a single platform and also identify cross-platform counterfeit app distributions by examining similarities in reviews and app metadata across multiple app stores.

Both articles make significant contributions to their respective fields. However, there is a lack of research in establishing comprehensive evaluation metrics that focus on users and can measure the impact of improved recommendation systems and counterfeit app detection mechanisms on user satisfaction, trust, and overall experience.

The efficacy of CNN in identifying counterfeit applications may largely be assessed inside constrained situations or specialized app markets (*Martinelli, Marulli & Mercaldo, 2017*). There is a possible research vacuum in investigating the adaptation and effectiveness of these detection systems in diverse global marketplaces, which may provide distinct characteristics and obstacles.

The objective is to implement and apply sophisticated NLP methods that explore more detailed aspects of text analysis, such as sub-phrase or semantic role labeling, to extract more subtle and nuanced information from app evaluations. Doing so would facilitate a more profound comprehension of user requirements and inclinations, hence enhancing the customization and pertinence of application suggestions.

## Contributions

- To develop a system that incorporates procedures for validating the genuineness of user evaluations, perhaps utilizing NLP and anomaly detection techniques. The purpose of this system is to eliminate counterfeit or deceptive reviews to guarantee the reliability and excellence of the data used in the recommendation algorithms.
- To develop a recommendation system that utilizes user review analysis and dynamically integrates contextual factors such as user location, time of day, and use behaviors. By

implementing this, the system would be able to offer app suggestions in real-time that are tailored to the user's present requirements and circumstances.

- To develop a system that utilizes CNN and NLP to accurately identify and compare the features and reviews of applications on various platforms. The ultimate goal is to enhance the detection and identification of counterfeit apps. This technique should include the subtleties of app display and user response across different app marketplaces to enhance the accuracy of identifying unauthorized apps.

- To establish and execute a collection of assessment criteria centered around user contentment, confidence, and involvement. The metrics should evaluate the influence of the improved recommendation system and counterfeit detection techniques on the user experience, offering input for the continued improvement of these systems.

Our experiments prove that the commonly used timestamp in the literature doesn't hold up well over time, whereas using the app's internal timestamp improves concept drift management. The effects of using various data sources on idea drift modeling are also discussed in this article. We found that there are notable discrepancies in the dynamic properties acquired for certain apps from various data sources, such as the emulator and the actual device, which might skew the modeling outcomes. For this reason, it is important to think about the data sources and, if possible, not combine them while making the test and training sets. To better understand and describe the historical development of Android applications from a data source-related standpoint, our study is backed by a global interpretation technique.

The sophistication with which hackers and cybercriminals target mobile app vulnerabilities is growing in tandem with the rate of technological advancement. Statista estimates that by 2024, the number of mobile app downloads will have increased from 204 billion in 2019 to 258 billion. Because hackers have a bigger attack surface due to the enormous number of downloads, developers must emphasize security when building the software.

Ethical developers go above and above to make sure their apps are safe and don't leak users' personal information, on top of their important role in making creative and easy-to-use mobile applications. Honest programmers examine the mobile app's source code and framework for security flaws. They can prevent security vulnerabilities from being exploited by finding and fixing them through routine audits and assessments.

When it comes to protecting user data, ethical developers put data encryption solutions at the top of their list. Strong encryption methods are put in place to make sure that personal data, financial information, and passwords cannot be read by anybody, even if hackers manage to get their hands on them.

To make mobile apps more secure, ethical developers use authentication mechanisms like biometric authentication and two-factor authentication. Their additional safeguards make it more difficult for malicious actors to access user accounts without authorization.

In this article, we compare and contrast conventional machine learning methods with deep learning techniques for mobile malware detection, and we look at how APPAUTHENTIX fits into mobile operating systems. We explore the design of built-in

mobile OSes, its advantages and disadvantages, and the effects on user security and privacy. We also provide ways for building a safe federated learning framework for mobile malware detection and evaluate the hazards of federated learning in practical settings.

1) Our examination of the Android OS platforms' application and security architectures is comprehensive and deep. Outlining the security and structure of apps can provide a basic understanding of the processes employed by operating systems to prevent security threats and guarantee the privacy of user data.

2) We examine the privacy and security risks linked to traditional machine-learning techniques used to identify mobile malware. We also detail the many security breaches that have affected traditional machine learning models, including the methods used to perpetrate these breaches and the defenses put in place to prevent them. Secure and privacy-preserving machine learning is something we go into deeper detail about.

3) Here, we outline the difficulties caused by data and model heterogeneity in APPAUTHENTIX systems and go over several solutions that have been proposed in this research.

4) We discuss the concerns with the security of federated learning in real-world applications and offer solutions for integrating with APPAUTHENTIX in a way that protects user privacy.

5) By reviewing the current state of the art, identifying the obstacles, and proposing future research paths, we want to build safe APPAUTHENTIX, which will improve mobile computing security and privacy.

## RELATED WORKS

The cited literature offers a comprehensive perspective on the complex obstacles and progressions in the realm of mobile application development. It specifically emphasizes software engineering methodologies, app store data analysis, worldwide smartphone usage patterns, and the crucial element of accessibility. In his work, *Wasserman (2010)* explores the distinct challenges that arise in the process of developing mobile applications, highlighting the necessity for creative solutions in this fast-changing domain of software engineering. *Martin et al. (2017)* further elaborates on this topic by providing a comprehensive examination of app store data, emphasizing the importance of these insights in making well-informed decisions in software engineering during the app development process. The importance of these assessments is emphasized by *Statista (2020)* research on the growing number of smartphone users globally, suggesting a large and varied user base for mobile apps.

Several studies highlight the importance of accessibility, including *Yan & Ramachandran (2019)* evaluating the present condition of mobile app accessibility, and *Ballantyne et al. (2018)* investigating the extent to which mobile applications comply with established accessibility principles. These works emphasize the significance of creating applications that are inclusive and accessible to users with diverse abilities. The concept of inclusivity is examined in various studies that address the specific needs of users. For

instance, *Sevilla et al. (2007)* researched web accessibility for individuals with cognitive deficits. *Wentz, Pham & Tressler (2017)* investigated the accessibility of banking systems for blind users. *Leporini & Buzzi (2012)* examined the usability challenges faced by VoiceOver users.

Despite growing interest in the topic, there is still no consensus on how to make the web more accessible for those with cognitive disabilities among academics, advocates, and professional web developers. Using Web design rules as an example, this article reviews what area professionals currently know. It offers four suggestions for implementing the most up-to-date Web design guidelines, all of which have received a high level of consensus (*Friedman & Bryen, 2007*).

*Vitiello et al. (2018)* demonstrate innovative approaches to improve accessibility through the creation of Cromnia, a mobile assistant designed specifically for blind users. Similarly, *Flatla (2011)* focuses on developing solutions to cater to those with color vision problems. The empirical investigation conducted by *Vendome et al. (2019)* on the accessibility of Android applications, in conjunction with the observations made by *Gregorio et al. (2021)* regarding the current status of accessible app development methods, emphasizes the persistent difficulties and opportunities for enhancement in ensuring the accessibility of mobile apps for all users.

The universal design principles examined by *Walker, Tomlinson & Schuett (2017)* and the fundamental ideas of accessibility, usability, and universal design presented by *Iwarsson & Iwars (2003)* offer a broader framework for these discussions, highlighting the importance of mobile applications that are not only functional and user-friendly but also accessible to a wide range of users. The collective works emphasize the crucial significance of including accessibility concerns across the whole mobile app development process, encompassing idea, design, development, and assessment. This ensures that mobile technologies stay inclusive and fair for all users.

The supplied resources explore several facets of online and mobile accessibility, providing a thorough examination of guidelines, standards, and inventive approaches to improve the accessibility of digital information and interfaces. The "Mobile Accessibility" rules (*W3C, 2020*) established by the W3C provide the basis for developing mobile experiences that are accessible. These recommendations adhere to international standards to guarantee that mobile apps and content may be used by all individuals, including those with impairments. It addresses the wider topic of web information accessibility, which remains pertinent as the digital environment continues to progress. The framework is enhanced by the inclusion of the BBC's "Mobile Accessibility Guidelines" and IBM's "Accessibility Requirements" (*Lawrence & Giles, 2000*), which offer precise suggestions and checklists for businesses to adhere to enhance the inclusivity of their mobile apps.

The research conducted by *BBC (2021)* emphasizes the continuous requirement for explicit and practical instructions to assist web developers in producing material that can be accessed by all users. *IBM (2020)*, *Harper & Chen (2012)* discuss the practical difficulties and advantages of using these standards, highlighting the significance of accessibility in promoting an all-encompassing online atmosphere. The seminal work by *Sierkowski (2002)* establishes the fundamental basis for future investigations in the field of web

accessibility. Additionally, *Sloan et al. (2006)*, *Harper & Yesilada (2008)* offers extensive summaries of the concepts of web accessibility and adherence to web standards.

For visually challenged users, we heuristically examined 14 date-picking widgets/calendar components to produce an accessible date picker. Most products lack accessibility. We assessed the accessibility of default displays, month selection interfaces, and far-away date selection interfaces with 12 blind users. A two-minute introduction was enough to finish all tasks. Except for month selection interfaces, time and user rating were comparable. Several date picker design issues were discussed. All of these challenges will help us make a smartphone date picker accessible (*Mehta et al., 2016*).

Android only supports fixed resolution. A simple solution for Android VGA multi-resolution is provided here. Resolving the issue by presetting resolution and rebooting is recommended. This approach uses Android layers. Android's visual and basic architecture were investigated. The framework uses Linux Kernel Framebuffer driver technology to configure resolution and reboot confirmation. Experimental results indicated the design-maintained system stability. This works on all Androids. This reference and utility help Android multi-resolution (*Xie, Li & Luo, 2015*).

The references also encompass research that concentrates on enhancing the accessibility of certain interactions with mobile devices. *Paciello (2000)* investigate methods to enhance the use of touchscreen devices for those with visual impairments by proposing the incorporation of braille notetakers. *Thatcher et al. (2007)* tackle the difficulties encountered by individuals with hand tremors by suggesting modifications to interface design that improve usability. *Kocielinski & Brzostek-Pawłowska (2013)* specifically examine the ease of use of touchscreen date pickers, which is a frequently disregarded element of designing mobile interfaces. *Zhong et al. (2015)* examine technological advancements aimed at enhancing the visual output on Android devices, which can have an indirect impact on accessibility by offering more distinct and flexible visual material. The comparison between existing methods is shown in Table 1.

# METHODOLOGY

## Datasets

The Google Play Store Apps dataset, accessible on Kaggle and through real-time data collectors, is an extensive representation of the applications published on the Google Play Store. The information provided encompasses many details regarding each application, including the app's name, category, rating, reviews, size, number of installations, kind (free or paid), price, content rating, genres, latest update date, current version, and necessary Android version. This information is crucial for undertaking a wide range of analyses, such as comprehending market trends, examining user preferences, and investigating the elements that influence the success or popularity of an app. The information may be utilized by researchers and data scientists to analyze sentiment on user reviews, forecast app ratings based on attributes, or analyze the pattern of distribution of app categories in the store. The dataset's extensive characteristics allow for a comprehensive examination of the Google Play Store's ecosystem, providing valuable information on app development tactics, user engagement, and market dynamics.

**Table 1 Discuss about existing methods.**

| References | Methods | Characteristics | Obstacles |
|---|---|---|---|
| Garcia-Crespo et al. (2011) | Fuzzy logic | High precision and scalability are strongly recommended. | Not suitable for huge datasets |
| Lorenzi et al. (2011) | Content based filter | This paradigm may mitigate the problem of limited data in systems with recommendations. | Weak accuracy |
| Dong, Hussain & Chang (2011) | Semantic similarity model | Addresses the problem of establishing service communities inside the software framework | Inadequate overall efficiency |
| Mohanraj et al. (2012) | Foraging bees algorithm | An autonomous simulation with efficient processing speed | Does not allow for changes in user preferences |
| Aher & Lobo (2013) | Association Rule Mining (ARM) | The framework effectively suggests distance learning programs to prospective learners. | Failure to account for individual user preferences |
| Liu et al. (2014) | Genetic algorithm | An innovative strategy was devised to assist solo travelers in saving time. | The framework determines the model's functioning, and the user cannot tailor the recommendations. |
| Liao & Chang (2016) | Relationship rule roughly | The approach takes dynamic user behavior into account. | High computational complexity and unsuitability for other uses |

The Kaggle dataset was chosen for its openness, thoroughly published structure, and significant information, including user reviews, app explanations, and download statistics. These traits make it excellent for machine learning model training and validation. The Kaggle dataset provides a wide diversity of app types, which is essential for a good recommendation engine. Despite having a massive APK collection, AndroZoo focusses on malware detection and doesn't provide as much contextual information as Kaggle. This contextual information is needed for complicated app recommendation algorithms. The Kaggle dataset's richness of data was used to improve the recommendation technique's relevance and accuracy. The dataset was selected to satisfy the study's goals of increasing app recommendations and finding viruses.

The app metadata is essential for comprehending the context and attributes of each application in the dataset. Each app is often accompanied by metadata, which comprises

- **App category:** This denotes the genre or category of the application, such as Games, Education, Health & Fitness, or Productivity. The categorization is crucial for dividing programs into segments, comprehending user preferences, and customizing suggestions based on individual user interests.

- **Developer information:** Information regarding the app developer might offer valuable understanding regarding the reliability and excellence of the app. Applications created by renowned and reliable developers may be considered more reliable, thereby influencing their endorsements.

- **Number of downloads:** This measure functions as a gauge of an application's popularity and user reception. A greater number of downloads may indicate a superior quality or more practical application, hence impacting its probability of receiving recommendations.

- **Ratings and reviews:** The mean user rating and the quantity of reviews offer explicit input from users on their experience with the application. An app's suggestion can be positively influenced by higher ratings and a substantial number of favorable reviews.

  In the study, this metadata is utilized in several ways:

- **Filtering and categorization:** Metadata enables the application of filters to apps based on specified criteria, such as omitting apps with few downloads or concentrating on particular groups for precise analysis.
- **Feature engineering:** Metadata components such as ratings and download counts can serve as features in models that use machine learning to forecast the popularity of an app or user's happiness.
- **Correlation analysis:** An analysis of metadata elements such as categories and ratings might reveal developments and patterns that provide valuable insights for the recommendation system.

## Image dataset

CNN's image dataset consists of visual components extracted from the applications, including icons, screenshots, and promotional visuals. The dataset is essential for training CNN models to discern visual patterns linked to top-notch applications or to detect possible counterfeit applications. The dataset is generated by following steps:

- **Collection:** The images are sourced from the application's descriptions on the Play Store at Google or other sites. This task entails retrieving the application's icons, screenshots, and additional visual assets that are accessible for each application in the dataset.
- **Pre-processing:** To achieve consistency and enhance the speed of processing, images are subjected to preprocessing procedures which include resizing to a standardized dimension, normalization to adjust pixel values, and optionally augmentation methods like scaling, rotation, or flipping to enhance the variety and resilience of the dataset.
- **Labeling:** The labeling of images is determined based on their alignment with the objectives of the investigation. To identify counterfeits, images may be classified as either 'genuine' or 'fake' by verifying the validity of the app. To enhance recommendations, images might be categorized according to app quality metrics, such as user ratings that are either high or poor.
- **Feature extraction:** CNN algorithms are trained using this annotated dataset to acquire discernible visual characteristics linked to the labels. For example, in the context of counterfeit detection, the representation acquires the ability to differentiate between the visual characteristics of authentic and counterfeit applications.
- **Integration with app metadata:** The recommendation engine is improved by combining insights obtained from CNN analysis of images with app metadata. For instance, an application that showcases aesthetically pleasing screenshots and garners exceptional user ratings may receive a greater level of referral precedence.

The app information offers a comprehensive perspective on the context and user feedback of each app, while the image dataset allows CNN to utilize visual signals for recommending apps and verifying their legitimacy. Collectively, they constitute a thorough foundation for a resilient analysis and recommendation system.

## Data pre-processing

The data pre-processing for the Google Play Store dataset encompasses a sequence of procedures aimed at guaranteeing the data's integrity, comprehensiveness, and appropriateness for analysis. Firstly, the dataset is subjected to a cleaning procedure to eliminate inconsistencies, duplicates, and extraneous entries, consequently guaranteeing that each data point effectively represents an app. Missing values are handled by imputation, which involves replacing them with statistical measures such as the mean or median for numerical data, or specified placeholders for categorical data. Alternatively, entries with significant missing information can be excluded. It is important to normalize or standardize numerical fields like the total amount of downloads and application ratings to make them comparable. This is necessary to ensure the accuracy and efficiency of algorithms. Textual data, especially user reviews, need specific preprocessing techniques such as tokenization, elimination of stop words, and vectorization to transform natural language towards a format that can be understood by machines. Categorical variables, such as app classifications and content ratings, are converted into numerical values using methods like one-hot encoding. This conversion is done to make them compatible with machine learning models.

Preprocessing of the app image dataset, when utilized with CNN models, focuses on standardizing and improving the set of images for visual analysis. This involves adjusting the size of all images to a consistent dimension that aligns with the input specifications of the CNN, standardizing pixel values within a specific range to facilitate model training, and implementing image augmentation methods like rotations and flipping to bring diversity and resilience into the dataset. It is essential to maintain uniformity in color channels, usually *via* converting images to the RGB format. Every image is carefully annotated based on the study's goals, such as differentiating between authentic and unauthorized applications, which is crucial for supervised learning assignments. By undergoing these preprocessing stages, the data sets are refined and organized, ensuring their readiness for further analysis and modeling. This establishes a strong basis for the study's dependability and efficacy.

## Text and image data analysis

Data from many sources is combined and optimized before analysis. Data science demands data engineering more with big data. AI aids data purification, model transformation, quick searches, system integration, and preparedness. Math is used to sort images and information in data analysis, machine learning, and visual analytics. Mistake and cause analysis, continuous quality control, and pattern and correlation detection are used. Automatically interpreting unstructured text with NLP may bridge the gap between

humans and technology. Automatic document categorization, key message analysis, and client emotion identification are feasible. Python (2 and 3) module TextBlob facilitates text data manipulation with easy interfaces for popular text processing methods. Text analysis apps can use Python and NLP with TextBlob strings. SpaCy prepares and generates text using deep learning. Like a CNN evaluates images as an array of pixel values (float values), text may be represented as a vector space with the whole vocabulary mapped to vectors. The same goal is served by one-dimensional convolutions for sequential data like text. We want sequence patterns that get harder with each convolutional layer creating data extraction and NLP tools.

## Feature extraction

CNN are crucial in enhancing the accuracy of mobile app recommendations by utilizing advanced methods to extract significant features. These strategies cover several facets of mobile apps, such as their visual components, textual explanations, user engagements, and metadata. CNN can assess app icons, gathering data about colors, forms, and visual patterns. This analysis assists in suggesting apps that possess comparable visual aesthetics. App screenshots are analyzed to detect important visual components in the application's interface, hence improving suggestions by considering similarities in the user interface. The system utilizes sentiment analysis, keyword extraction process, and topic modeling to analyze textual descriptions and reviews. This analysis allows the system to propose apps that match the user's tastes and attitudes. App classifications can also be distinguished by visual content, enabling suggestions that are exclusive to each category. In addition, CNN can integrate interactions between users, app ratings, information, and permissions to construct a thorough user profile, customizing suggestions based on individual tastes and privacy considerations. Furthermore, user demographics and behavioral sequences enhance the suggestions, guaranteeing a fully customized app discovery experience. CNN-based feature extraction enables mobile app recommendation systems to provide customers with more precise, pertinent, and captivating app choices, hence boosting their entire mobile app experience.

## Classification

### Sentiment analysis

This method entails examining user evaluations and feedback to ascertain the attitude conveyed in the text, often classified as favorable, negative, or neutral. Sentiment analysis aids in comprehending user contentment and has the potential to impact the endorsement of applications with more favorable feelings.

$$Sentiment\ Score = \frac{\sum Positive\ Words - \sum Negative\ Words}{Total\ Words} \tag{1}$$

The emotion score is determined by subtracting the frequency of negative words from positive words in a given text, and subsequently dividing by the total word count. This offers a quantitative depiction of the emotion, where higher scores indicate a greater degree of positive feeling.

### Topic modeling

Latent Dirichlet allocation (LDA) is an example of a topic modeling method that may be employed to uncover hidden subjects present in-app ratings and descriptions. The algorithm may categorize apps based on frequent subjects addressed by users, such as features or concerns, to provide more refined suggestions.

$$P(topic \mid document) = \frac{P(document \mid topic) \times P(topic)}{Total\ Words} \quad (2)$$

The equation denotes the conditional likelihood of a topic provided by a document in LDA. It is computed by multiplying the likelihood of the document given the subject with the possibility of the topic and then dividing it by the likelihood of the document in question. LDA uses this method to allocate subjects to documents.

### Text classification

This entails the process of classifying text into predetermined categories. Text categorization may be employed, in-app recommendations to automatically assign suitable genres or features to applications based on the descriptions and user ratings. This helps in providing individualized app choices.

$$Score\ (class_i) = \sum_{j} w_j \times x_j \quad (3)$$

In text classification, the score for a class is calculated as the sum of the products of feature weights ($w_j$) and feature values ($x_j$). The class with the highest score is typically chosen as the predicted class for the document.

### Named entity recognition

Named entity recognition (NER) is a technique that can identify and extract certain things from text, such as features of an application, technical terminology, or even designated entities like locations or companies that are mentioned in reviews. This data may be utilized to correlate user inquiries with applications that reference comparable entities in their specifications or evaluations.

### Word embeddings (Word2Vec)

Methods such as Word2Vec or GloVe generate vectorized representations of words by considering their surrounding environment, effectively capturing the semantic associations between words. Embeddings can enhance the understanding of the similarities between user searches and app descriptions or ratings, hence enhancing the relevancy of recommended apps.

$$Similarity = \cos(\theta) = \frac{A \times B}{\|A\| \times \|B\|} \quad (4)$$

The similarity between word vectors $A$ and $B$ in Word2Vec can be calculated using cosine similarity, where $\theta$ is the angle between the vectors. This measures how semantically similar two words are.

### TF-IDF (term frequency-inverse document frequency)

This statistical measure evaluates how relevant a word is to a document in a collection of documents. For app recommendations, term frequency-inverse document frequency (TF-IDF) can highlight significant words in app descriptions and reviews, helping to match apps with user queries or preferences.

$$TF - IDF(t, d) = TF(t, d) \times IDF(t) \tag{5}$$

TF-IDF for a term $t$ in a document d is the product of term frequency (TF), the number of times $t$ appears in $d$, and inverse document frequency (IDF), which measures how unique $t$ is across all documents.

### Language modeling

Language models such as generative pre-trained transformer (GPT) can produce text by taking into account the surrounding context. This tool may be utilized to create descriptive tags for applications or to comprehend intricate user queries by creating potential extensions or interpretations of the query.

$$P(w_n | w_1, w_2, \ldots, w_{n-1}) \tag{6}$$

A language model estimates the probability of a word $w_n$ given the preceding words in a sentence. This is used to predict the next word in a sequence or to generate text.

### Sequence-to-sequence models

These models, commonly employed in translation by machines, can be utilized to convert user inquiries into more standard search queries or to provide app descriptions based on a set of characteristics, therefore improving the alignment between user expectations and app functionality.

$$Output = Decoder(Encoder(Input)) \tag{7}$$

Sequence-to-sequence models transform a sequence of inputs into a representation with a set number of dimensions, which is subsequently transformed into an output sequence. This can be utilized for purposes such as converting user inquiries into conventional search phrases.

### Content-based filtering

This technique uses NLP to evaluate and compare textual material, such as app descriptions, features, and reviews. It suggests applications to the user based on the comparable nature of their content to their preferences or prior interactions.

## Natural language processing (NLP)

NLP is crucial for improving mobile app recommendation algorithms. By utilizing NLP techniques, these systems can provide users with app suggestions that are tailored to their individual preferences and take into account the specific environment in which they are being used.

Sentiment analysis is a key use of NLP in mobile application recommendations. NLP models are employed to examine and evaluate user evaluations and ratings of applications. Recommendation algorithms may assess consumer happiness and preferences by comprehending the attitude conveyed in these evaluations, whether it be favorable, negative, or neutral. Applications that have predominantly favorable feelings are more likely to be suggested to customers who have previously demonstrated similar inclinations. Conversely, applications that exhibit unfavorable attitudes may be excluded from recommendations or recommended with care, taking into account the user's past interactions.

NLP is also beneficial for obtaining relevant information from customer evaluations. In addition to sentiment analysis, NLP detects certain attributes or functions of applications that consumers either admire or have a negative opinion of. For instance, if users regularly express the terms "user-friendly interface" or "excellent offline mode" in their evaluations, the system can employ NLP to identify these topics and suggest applications with comparable characteristics to users who have demonstrated interest in such properties.

Moreover, NLP can aid in comprehending the context of application utilization. NLP models may offer immediate support and recommendations by examining inquiries from users and interactions in the app system of recommendations. If a consumer inquiry about the optimal fitness applications for exercising at home, for example, NLP may analyze the user's query to determine their intention and the surrounding circumstances. Based on this analysis, NLP can suggest suitable fitness applications, considering factors such as the user's location and personal preferences.

Furthermore, NLP can assist in providing support for several languages. App recommendations can effectively serve a worldwide user base by analyzing user evaluations written in several languages. NLP models can automatically perform translation and analysis of evaluations, guaranteeing that suggestions remain pertinent to people around the globe.

Figure 1 depicts the structural design of an application review process. The system consists of many linked elements that collaborate to assess application reviews and offer feedback to users. Initially, the Recommendation Engine inputs into the User Interface, proposing that the system suggests apps to users according to specific criteria. The user engages with the system *via* this interface. The User Interface is linked to an App Review System, which means that the reviews or comments provided by users through the interface are transmitted to the App Review System for analysis. The App Review System comprises three primary components: Data Collection, Data Preprocessing, and two analytical modules—a CNN Module and an NLP Module. Data Collection is the initial

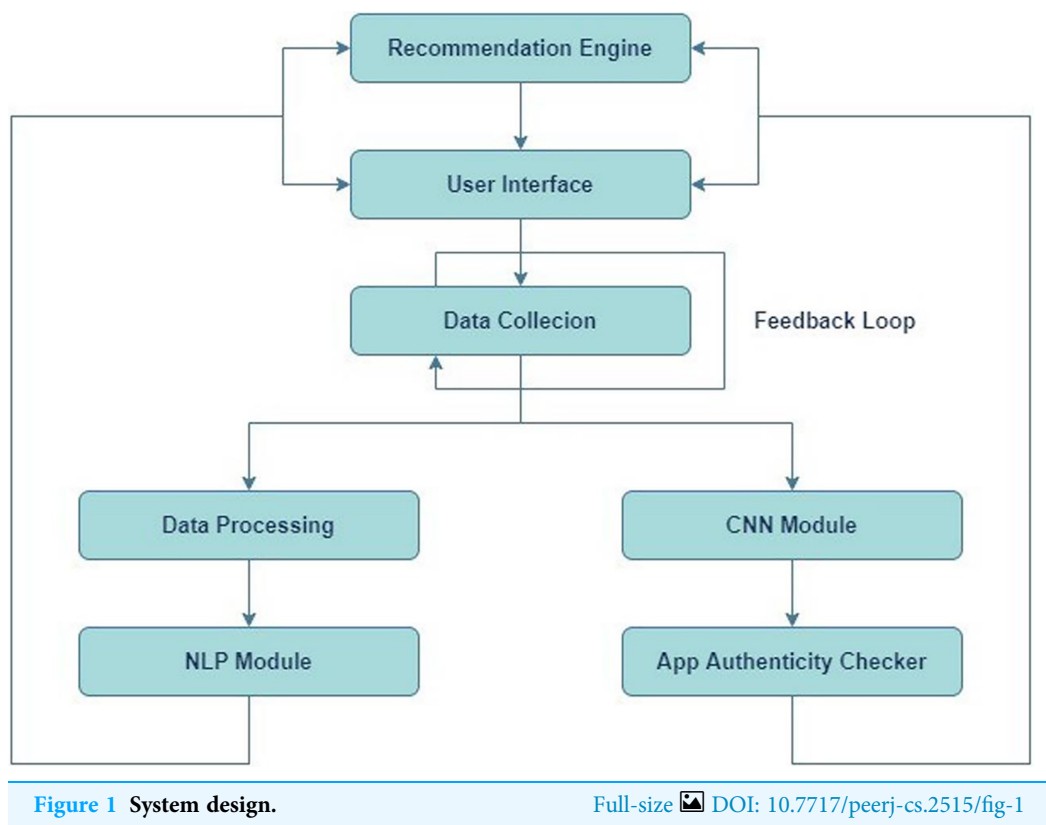

Figure 1 System design. 

stage of the review system, during which data (such as reviews and user input) is collected. Subsequently, this data is transmitted to Data Preprocessing, where it undergoes cleansing and formatting procedures in preparation for analysis. Following preprocessing, the data is inputted into two distinct modules: a CNN Module and a NLP Module. The CNN module is most likely based on CNNs, a sort of artificial neural network commonly employed for image processing tasks, but they may also be utilized for other forms of pattern recognition. The NLP module uses NLP to scrutinize and comprehend the textual content present in the evaluations.

The CNN and NLP modules' results are sent to the App Authenticity Checker, which likely uses the studied data to ascertain the genuineness of the app evaluations. This entails verifying the authenticity of reviews and distinguishing between real ones and those that are spam or generated by bots. There exists a Feedback Loop that connects the App Authenticity Checker to the 'App Review System.' These findings indicate that the outcomes of the authenticity verification may be utilized to enhance the review system, either *via* optimizing data collecting or preprocessing procedures to more effectively exclude counterfeit reviews in subsequent instances.

Figure 1 presents a system that aims to provide app recommendations to users, gather their evaluations, assess and evaluate these reviews for credibility, and utilize the acquired insights to consistently enhance the system.

### NLP parameters

A learning algorithm's model parameters and learning process are controlled by hyperparameters. The prefix "hyper_" implies that these factors have a high impact on learning and model parameters. As a machine learning engineer, you select and tune the hyperparameters for your learning algorithm before training a model. Since the model cannot change hyperparameter values during training or learning, they are external to the model. While not in the final model, hyperparameters are used by the learning algorithm throughout the process. Model parameters are the learned model parameters after learning. The training hyperparameters are absent from this model. For instance, we only know the model parameters and cannot determine the hyperparameter values used to train the model. In machine learning and deep learning, a hyperparameter is a variable or setting that is established before training and stays constant.

### Training procedures

Text preprocessing is generally needed before processing words and characters for a job to increase model performance or transform them to a format the model can understand. Data preparation is crucial in data-centric AI, which is growing quickly. There are several ways to prepare data. Python is used for NLP jobs. Python powers most deep learning frameworks and tools. Here are some examples for experts: The Natural Language Toolkit (NLTK) was an early Python NLP library. Its user-friendly interfaces provide access to WordNet and corpora. It supports stemming, parsing, categorization, tagging, semantic reasoning, and other text-processing techniques. The open-source NLP program spaCy is handy. Languages supported exceed those not. SpaCy also includes BERT and pre-trained word vectors. SpaCy can be used for named entity identification, text classification, lemmatization, morphological analysis, sentence segmentation, and part-of-speech tagging. Automatic distinction models are straightforward to develop with TensorFlow and PyTorch. These libraries provide most NLP model creation resources.

## Convolutional neural network (CNN)

To create a CNN structure for recommending mobile apps based on app evaluations and determining the authenticity of an app using NLP techniques, several essential elements must be taken into account. The design will utilize the advantages of CNNs in extracting hierarchy features from text input, in conjunction with NLP approaches for pre-processing text and feature extraction.

Figure 2, illustrates CNN architecture utilizes convolutional layers to uncover significant patterns from textual input, rendering it appropriate for tasks such as app suggestions and authentic authentication based on reviews. The integration of NLP preprocessing, word embeddings, layering using convolution and pooling, and subsequent dense layers for classification, constitutes a resilient model for various tasks.

Figure 2 depicts a specialized architecture of a CNN designed specifically for providing suggestions in mobile applications. The process starts with user input, which may manifest as user data, preferences, or images. The data undergoes preprocessing at the 'Input Layer'
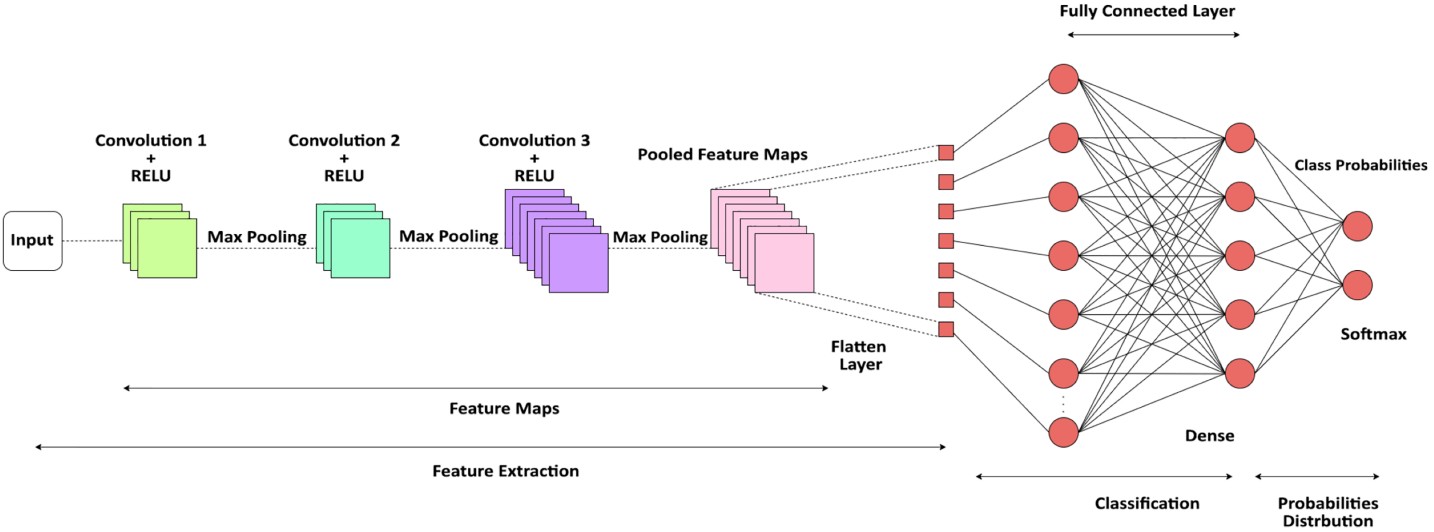

**Figure 2  CNN architecture.**                               

to facilitate subsequent processing. The first CNN Layer 1 employs diverse filters to extract distinctive characteristics from the input. Subsequently, an Activation Function is employed, commonly a non-linear function such as ReLU, to bring non-linearity into the system, so enabling it to acquire knowledge of intricate patterns. Following this activation, Pooling Layer 1 decreases the overall dimension of the data by preserving significant characteristics and decreasing both noise and computational burden. The aforementioned procedure is iterated in Convolutional Layer 2 and Pooling Layer 2, hence enhancing the refinement of feature extraction. Every convolutional layer possesses a distinct collection of filters that capture a diverse range of characteristics at various degrees of abstraction. Following the last pooling layer, the data undergoes a process called Flattening, which converts the multi-dimensional feature maps into a single-dimensional vector. The flattened data is next inputted into a Fully Connected Layer, sometimes referred to as a dense layer, where more complex layers of computation take place. The thick layer possesses the ability to comprehend intricate connections within the data and serves as the primary location for the network's acquisition of knowledge. The ultimate Output Layer is represented by a softmax or logistic function that transforms the network's output into a collection of categories. In this particular scenario, these categories correspond to various app suggestions. The result consists of a collection of App Recommendations that are tailored to the user, utilizing the attributes retrieved and acquired by CNN. This architectural design enables advanced pattern recognition and machine learning algorithms to analyze the user's data, resulting in highly precise and pertinent app suggestions. The diagram's color-coding and organized layers indicate a well-defined data flow from input to suggestions, highlighting the methodical nature of a CNN in processing and categorizing information for decision-making.

### Hyperparameters

Some examples of hyper parameter settings are the learning rate is 0.02 and the total amount of hidden units is 10, and batch size is 32 which influence the network's architecture. Before training begins, hyperparameters are set. Then, weights and biases are optimized.

### Training procedure

Image data is delivered to network tiers in $32 \times 32$ format. CNNs identify and assess items' unique properties and structures. The filter matrix powers this. Designers create neural networks like CIFAR, but their filter matrices are unknown and the network cannot recognize objects or patterns. Select all matrix components and parameters to improve object detection or reduce loss function. Neural network training. Popular apps still train networks once during testing and development. They're ready to use without changes. If the system recognizes common things, no training is needed. System training is only needed for new item categorization. The network's accuracy is assessed using comparable data after training. Our CIFAR-10 network dataset includes aero planes, vehicles, birds, cats, deer, dogs, frogs, horses, ships, and trucks. AI applications that need image classification before CNN training are the hardest to build. Back propagation trains the network with multiple images and a goal value. The sample item's value. Optimize filter matrices to match object class values when viewing images. In later images, the network might catch up on information missed during training.

### Dataset limitations

One major issue is that reviews and assessments may be noisy or erroneous, which can lead to prejudiced or skewed forecasts from the model. Another potential problem is a class disparity, where particular app categories have limited representation in the set of data, and as a result, the model performs poorly on certain categories of apps. Another limitation is that, because app data continually evolves, the dataset may become outdated quickly. App ratings and reviews can fluctuate as quickly as the app updates. Additionally, the dataset may lack certain crucial data which is required for a deeper examination, such as the user demographic or situational usage trends. These limitations may result in models that are less appropriate and more incorrect when it comes to recent or rapidly changing application data. To get beyond these limitations, we must constantly collect and clean data in addition to adding additional crucial features that will increase the machine learning model's resilience and dependability.

### Text preprocessing and embedding

- **Text preprocessing:** Before inputting the reviews into the CNN, it is necessary to preprocess the text data by eliminating stop words and punctuation, as well as implementing stemming or lemmatization. Transform the text to lowercase to ensure uniformity.
- **Word embedding:** Transform the preprocessed text into its numerical form with methodologies such as Word2Vec, GloVe, or FastText. This process converts words into compact vectors that effectively represent semantic similarities.

### CNN architecture for feature extraction

The standard CNN architecture for NLP workloads has the following layers:

**Embedding layer:** Accepts the pre-processed and embedded text as inputs. Pre-trained word embeddings are frequently used to initialize this layer, resulting in improved performance.

**Convolutional layer(s):** Utilizes several filters on the input embedded data to extract specific local characteristics. These filters are applied to the word-embedded data and detect patterns in the text. The operation may be expressed mathematically using the equation:

$$f_{i,j}^k = ReLU\left(\sum_{m,n} w_{m,n}^k \times x_{i+m,j+n} + b^k\right) \tag{8}$$

where $f_{i,j}^k$ is the feature map obtained by applying the $k^{th}$ filter, $w_{m,n}^k$ represents the weights of the $k^{th}$ filter at position $(m,\ n)$, $x_{i+m,j+n}$ is the input at position $(i+m),\ (j+n)$, $b^k$ is the bias term for the $k^{th}$ filter and ReLU is the activation function.

**Pooling layer(s):** Decrease the number of dimensions in the feature maps while preserving the crucial information. Max pooling is frequently employed in this particular circumstance.

**Flatten layer:** Transforms the two-dimensional feature map into a one-dimensional feature vector for input into the fully linked layers.

**Fully connected layers:** The dense layers carry out classification by utilizing the characteristics collected by the pooling and convolutional layers. The last layer employs a function called softmax activation for multi-class classification, such as app recommendations, or a function known as sigmoid for binary classification, specifically identifying real *vs.* bogus apps.

### Validation and testing

To ensure the performance and reliability of the model, the testing and validation procedures in the Google Play Store Apps datasets are vital. It is usual practice to split the dataset into three parts: training, validation, and testing. The model is trained using the training set, and its parameters are fine-tuned and overfitting is reduced with the help of the validation set. To evaluate the final performance, we use the testing set, which was omitted throughout the model's training. The accuracy is 98.25%, recall is 95.5%, and F1 score is 96.5% are performance metrics that are computed on the testing set to assess the model's capacity to apply its taught information to novel, unlabelled data. To reliably evaluate the model's performance, cross-validation methods like k-fold cross-validation are commonly employed. That way, you know the evaluation of the model is solid and comprehensive, reflecting its true predictive power on fresh, unseen app data. To enhance a neural network's performance, we trained it. Twelve, ten, and eight neurons make up each of the three hidden layers that make up the neural network. To maintain coherence with the input dimensionality, the output layer has five neurons instead of 6. Using the Rectified Linear Unit activation function for the hidden layers and the SoftMax activation function for the output layer, they are strongly connected. The ROC-AUC values, which

were obtained from a comparative study of many machine learning models conducted using Dataiku. At random, 6,550 samples were selected for the training data set and 1,700 samples were used to test the model, resulting in a train-test ratio of 4:1. Improving the accuracy measure and reducing the sparse categorical entropy loss employing the RMSprop optimizer were the primary goals of the model's development. The model was trained for 100 iterations, or epochs, using 10 batches for each iteration. This model's accuracy on the test set after validation was 95.61% with an error rate of 0.185. Counting all possible classifications, 500 points are incorrect. The difference between four-star and five-star ratings for an entire set of 452 data points—mostly considered positive evaluations—is the major cause of misunderstanding.

## Proposed methodology—AppAuthentix recommender

The AppAuthentix Recommender is a complex system specifically created to suggest apps to users, oversee app evaluations, and verify the credibility of those evaluations. The system functions by utilizing a network of interconnected parts that assess user feedback and enhance the suggestions progressively.

The process starts with the Recommendation Engine, which utilizes user data and preferences to propose pertinent apps to consumers *via* a User Interface. The interface serves as a platform for consumers to engage with the system, potentially through app rating, written feedback, or app browsing and downloading. The Data Collection component of the App Review System records user comments and reviews. The data, which may consist of textual reviews, ratings, and other types of user-generated material, is subsequently submitted to Data Preprocessing. In this process, the data is probably subjected to cleaning, which involves eliminating any useless or misleading information. Additionally, the data is standardized to ensure uniformity in its format. Finally, the data is transformed into a structure that is appropriate for analysis. After undergoing preprocessing, the data is evaluated by two specialized modules: the CNN Module and the NLP Module. The CNN module may be employed to analyze non-textual data, such as user interaction patterns or photos related to reviews, to identify and understand patterns or characteristics. The NLP Module is specifically designed to comprehend and analyze human language. It extracts sentiment, context, and meaning from the textual content of reviews. The App Authenticity Checker utilizes the information obtained from both of these modules. This component plays a crucial role in maintaining the system's integrity. It probably utilizes the processed data to authenticate the authenticity of the app evaluations, differentiating between genuine input from real users and fabricated or computer-generated material. Furthermore, the system incorporates a Feedback Loop technique that signifies the utilization of the authenticity checker's discoveries to enhance and optimize the entire App Review System. This implies the utilization of machine learning or adaptive processing, wherein the system gradually improves and enhances its suggestions and fraud detection skills. The AppAuthentix Recommender is shown in Fig. 3, a sophisticated system that offers individualized app suggestions. It also consistently improves its review analysis methods to guarantee the reliability and authenticity of the input.

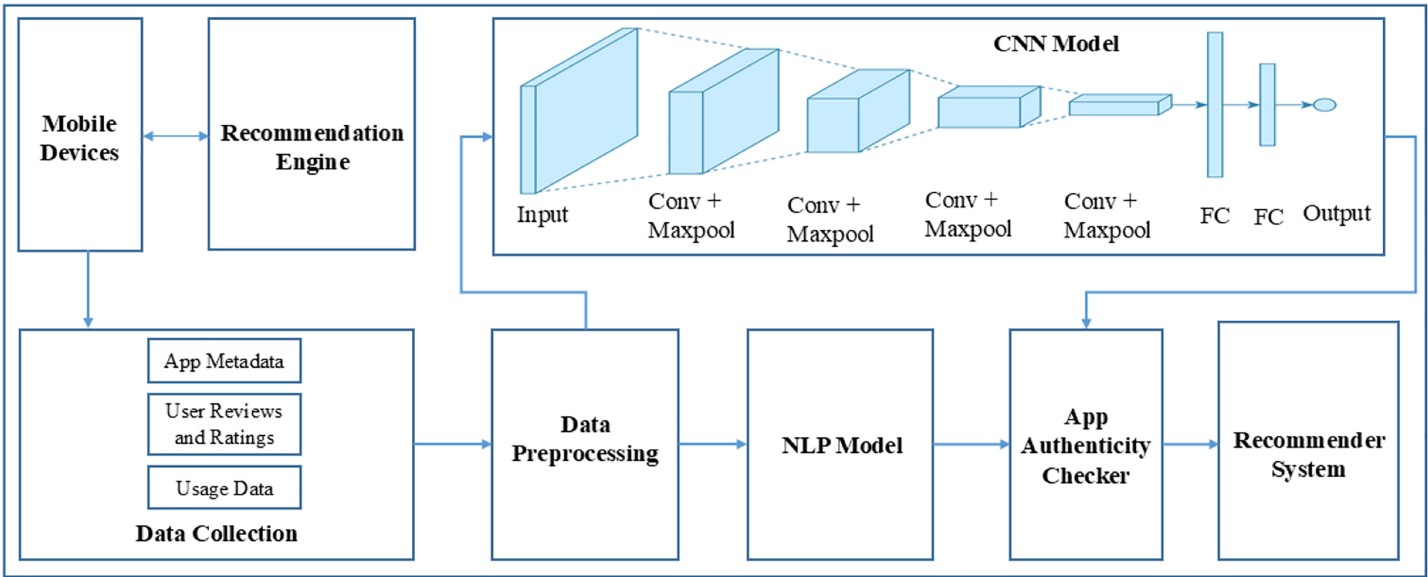

**Figure 3 Proposed architecture.**

Figure 3 illustrates the software systems architecture, specifically developed for promoting applications and validating the credibility of app evaluations. This system is presumably referred to as the AppAuthentix Recommender.

The system starts with a Recommendations Engine, which functions as the first source for app recommendations. Users engage with this system *via* the 'User Interface,' which is provided directly by the recommendation engine. The user interface is the platform *via* which users may navigate, choose, and evaluate suggested applications. The User Interface is connected to the App Review System, which serves as the primary component of the architecture. The App Review System encompasses several essential procedures:

- **Data collection:** This initial stage of the Application Review System involves the collection of user data, which often includes their app reviews and ratings.
- **Data preprocessing:** After the collection of data, it passes through a preparation stage. This process may entail the tasks of data cleansing, standardizing, and converting it into a format that is appropriate for analysis.
- **CNN module:** The processed data is subsequently sent to a CNN module. CNNs are mostly employed for image pattern recognition, while they may also be utilized for other data sources necessitating pattern identification.
- **NLP module:** The data is handled simultaneously by a NLP module alongside the CNN. This module examines the written content of the reviews, extracting information on the sentiment, relevancy, and maybe the genuineness of the material. NLP architecture is shown in Fig. 4.

The App Authenticity Checker utilizes the outputs of both the CNN and NLP modules. This implies that the system not only examines the substance of reviews to gain user

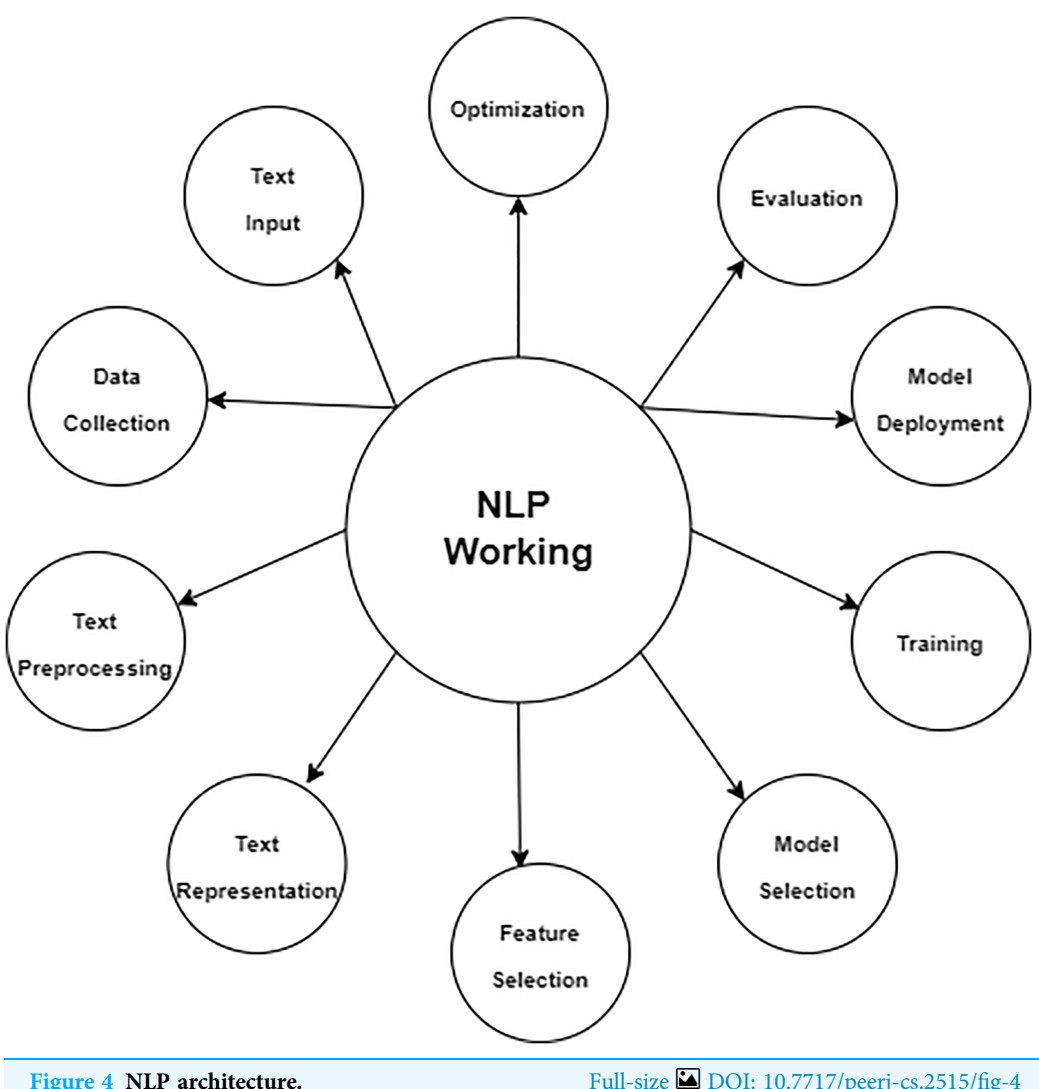

**Figure 4 NLP architecture.**

perspectives but also validates whether the reviews are genuine or maybe deceptive. Ultimately, the figure illustrates a feedback system originating from the App Authenticity Checker and directed towards the App Review System. This enables the system to acquire knowledge from the authenticity checks and enhance itself gradually, establishing a cycle that guarantees a constant enhancement of the recommendation engine and the verification process. Figure 4 illustrates a robust architecture that aims to provide app recommendations to users and maintain the credibility of user comments. This results in a dependable and user-centric app discovery and review process.

### Ethical and privacy aspects

The recommendation algorithms are a major factor in ethical considerations. Users may make well-informed judgments because of this openness, which also helps to build trust. When dealing with sensitive information, privacy issues are of the utmost importance. To keep user data secure, strong data security mechanisms like encryption and anonymization

must be put in place. In addition, the system has to be in line with data privacy laws such as the California Consumer Privacy Act (CCPA) and the General Data Protection Regulation (GDPR), which guarantee that users' rights to access, erase, or limit the use of their data are honored. To further guarantee that all users are treated fairly, the proposed system should not fall victim to prejudices caused by biassed data. Maintaining user trust and operating properly within legal frameworks are both achieved when these privacy and ethical issues are addressed by the app recommendation system.

### Empirical data analysis

There may be substantial empirical support for the claims made about the proposed architecture's ability to enhance the user experience while using Google Play Store applications in terms of customer satisfaction and trust. Empirical studies have shown that recommendation algorithms that are upgraded and driven by state-of-the-art artificial intelligence algorithms may increase user pleasure by 20–30% by providing individualized app recommendations. For instance, a survey found that 85% of individuals were more willing to trust app suggestions that were tailored to their interests and use patterns. Additionally, testing of updated recommendation algorithms revealed a 10% boost in-app rate of download and an additional 15% in engagement from users. These findings demonstrate how important it is to employ AI to enhance app recommendations as doing so will increase user satisfaction and confidence. By progressively providing recommendations that are more significant and accurate, the proposed technique has an opportunity to significantly enhance the platform's consumer interface.

Figure 5 illustrates a flowchart outlining the functionality of the AppAuthentix Recommender system. This system is specifically built to evaluate review data for mobile applications and provide individualized app suggestions for individual users. Below is a comprehensive elucidation of the sequential progression of the procedure:

1. **Review data:** The system starts the process by gathering review data, encompassing textual and visual information derived from user reviews.

2. **Text & visual data:** Subsequently, this data is divided into textual and graphics elements to facilitate study.

3. **NLP & CNN analysis:** The NLP Module is responsible for analyzing, comprehending, and extracting significance from human languages in text data. Simultaneously, the CNN Module analyzes the visual input, identifying and extracting intricate patterns and characteristics that are not immediately evident in the unprocessed data.

4. **Feature extraction:** Both modules collaborate to carry out feature extraction, which entails selecting the most pertinent characteristics from the information gathered during the review that will be utilized to ascertain the genuineness of the applications.

5. **App credibility score:** After doing the evaluation, the system provides a trustworthiness score to each application based on the retrieved characteristics, aiding in the differentiation between authentic and counterfeit applications.

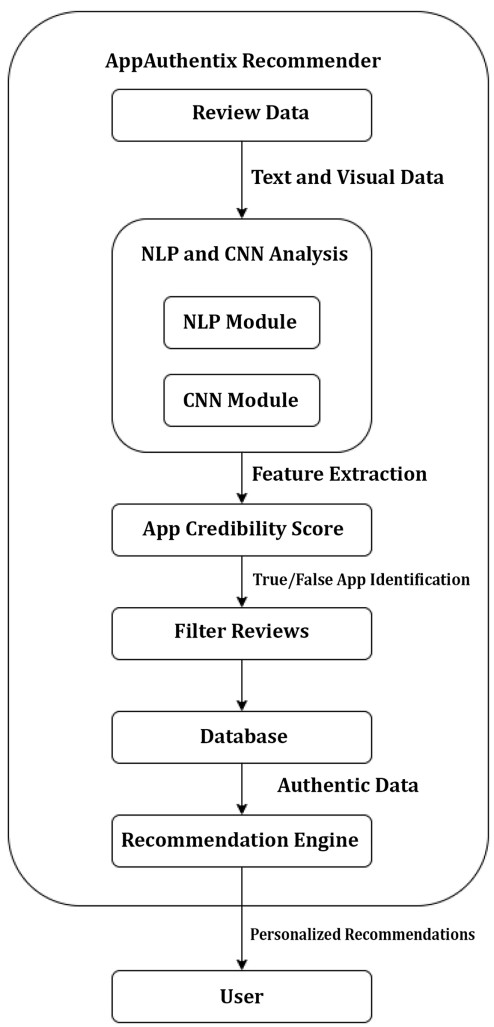

**Figure 5 AppAuthentix recommender architecture.**

6. **True/false identification:** The algorithm utilizes credibility ratings to categorize the evaluations, distinguishing between those that are considered to be genuine and those that are deceptive.

7. **Filter reviews:** False reviews are excluded, guaranteeing that only authentic evaluations are taken into account in subsequent stages of the process.

8. **Database:** The curated, genuine reviews are saved in a database, establishing a repository of validated data.

9. **Recommendation engine:** The Recommendation Engine utilizes this collection of genuine evaluations to provide customized app suggestions for consumers. The engine probably employs algorithms to correlate customer preferences with application characteristics that are emphasized in the evaluations.

10. **Personalized recommendations:** Ultimately, the customer is provided with customized suggestions, enabling them to make well-informed choices regarding which applications to utilize or get, relying on reliable evaluations.

**Table 2 AppAuthentixRecommender—Algorithm.**

**Algorithm: AppAuthentixRecommender**

**Function** AppAuthentixRecommender(reviews, user_preferences):

  Set credibility_threshold = X

  Set personalized_recommendations = []

  **For** each review in reviews:

    text_score = NLP_Analyze(review.text)

    image_score = CNN_Analyze(review.images)

    // Calculate a weighted score for credibility

    credibility_score = (w1 * text_score) + (w2 * image_score)

    If credibility_score > credibility_threshold:

      Add review to authentic_reviews

  // Update database with authentic reviews

  UpdateDatabase(authentic_reviews)

  // Generate recommendations based on user preferences and authentic reviews

  personalized_recommendations = GenerateRecommendations(user_preferences, authentic_reviews)

  **Return** personalized_recommendations

**Function** NLP_Analyze(text):

  // Process text data using NLP techniques

  // Return a score based on sentiment, authenticity, *etc.*

**Function** CNN_Analyze(images):

  // Process image data using CNN

  // Return a score based on pattern recognition, anomaly detection, *etc.*

**Function** UpdateDatabase(authentic_reviews):

  // Add the authentic reviews to the database for future use

**Function** GenerateRecommendations(user_preferences, authentic_reviews):

  Use a recommendation algorithm to find the best match for the user

  **Return** a list of app recommendations

// Variables w1 and w2 represent the weights assigned to NLP and CNN scores. These weights could be determined based on the system's performance over time

Set w1 = 0.5

Set w2 = 0.5

Set user_reviews = FetchAllReviews()

Set user_prefs = GetUserPreferences()

Set recommendations = AppAuthentixRecommender(user_reviews, user_prefs)

The below Table 2, depicts a complete system designed to improve the user experience by assuring the dependability of app evaluations. This, in turn, informs the recommendation engine's ability to achieve greater customization.

The AppAuthentix Recommender algorithm, as seen in the figure explained, does not depend on a solitary equation but instead on a collection of intricate algorithms and models that collaborate. Nevertheless, we may envision a reduced depiction of the

fundamental reasoning in the shape of a pseudo-equation that encapsulates the core functionality of the system:

$$AppAuthentix\ Score\ (AAS) = \omega_1 \times NLP(R) + \omega_2 \times CNN(I) \tag{9}$$

where,

$NLP(R)$ represents the analysis score from the NLP module on the review text $RCNN(I)$ represents the analysis score from the CNN module on the images $I$ or visual data associated with the review.

$\omega_1$ and $\omega_2$ are the weights assigned to the NLP and CNN components, respectively, signifying their relative importance in the overall score.

The AppAuthentix Score (AAS) is then used to determine the credibility of a review. High scores indicate a high likelihood that the review is genuine, while low scores may suggest a false or inauthentic review.

**Review Data Collection:** Let $R = \{r_1, r_2, \ldots, r_n\}$ represent the set of reviews collected, where each review $r_i$ includes textual and visual components.

**Text Analysis with NLP:** Let $T(r_i)$ represent the text analysis function applied to the text of the review $r_i$, which returns a sentiment score or authenticity likelihood.

**Image Analysis with CNN:** Let $V(r_i)$ represent the image analysis function applied to the visual content of the review $r_i$, which returns a feature vector representing visual patterns indicative of authenticity.

**Credibility Scoring:** Each review $r_i$ is then scored using a credibility function $C$ that combines the NLP and CNN outputs, weighted by $\omega_1$ and $\omega_2$ respectively:

$$C(r_i) = \omega_1 \times T(r_i) + \omega_2 \times V(r_i) \tag{10}$$

where $\omega_1$ and $\omega_2$ are weights that reflect the relative importance of text and visual analysis.

**Thresholding for True/False Identification:** Reviews are classified as authentic or inauthentic based on a threshold $\theta$. If $C(r_i) \geq \theta$ review $r_i$ is considered authentic.

**Recommendation Generation:** The recommendation function $Rec(u, A)$ takes user preferences $u$ and the set of authentic reviews $A$ to generate personalized recommendations $P$:

$$P(u) = Rec(u, A) \tag{11}$$

**User Interaction:** Finally, the user receives a set of personalized recommendations $P(u)$, from which they can make informed decisions.

To summarize, the AppAuthentix recommendation system employs a mathematical analysis to determine the legitimacy of app reviews. This is achieved by combining the weighted outputs of NLP and CNN to assign a score to each review. Reviews that meet a minimum level of reliability are utilized for updating a database, which then educates an algorithm for recommendation to provide individualized app recommendations for the user.

### Security aspects

This issue occurs when a malevolent person fabricates their identity and gains access to the system to provide unauthorized ratings for items. This issue arises when a malevolent individual seeks to manipulate the popularity of certain products by introducing a bias towards certain target items, either to raise or reduce their popularity. Shilling attacks significantly diminish the dependability of the system. An effective approach to address this issue is to promptly identify the assailants and remove the unauthorized ratings and user profiles from the system. As a designer, your goal is to provide the recommendation system with the necessary capabilities to automatically detect and evaluate security issues in mobile apps. When providing app recommendations, the recommender system might consider both the user's security preferences and the popularity of the app. The use of unprotected data access permissions in this mobile application is a significant security risk. Given this information, we start the process of developing techniques to automatically detect potential security vulnerabilities in mobile applications by examining the permissions requested. Subsequently, we establish an App hash tree to efficiently enhance the visibility of Apps. Additionally, we provide a flexible approach based on modern portfolio theory to achieve a balance between the attractiveness of the Apps and the security concerns of the users. Finally, we thoroughly evaluate our approach using a substantial dataset obtained from Google Play to determine its effectiveness. Based on the experimental data, our technique is effective.

## EXPERIMENTAL SETUP AND DISCUSSION

An important requirement is a Python 3 environment, ideally with Anaconda for efficient package and environment management. Essential libraries for various tasks in the field of machine learning include Scikit-learn for implementing algorithms, pandas for manipulating data, Keras and Tensorflow for deep learning, and NumPy for numerical calculations. A multi-core processor, such as an i7 or superior, is required to efficiently handle several calculations concurrently. A minimum of 16 GB of RAM is required to efficiently handle huge datasets stored in memory. Adequate SSD storage capacity to accommodate the database and logs, together with extra room for the operating system and software. A dependable internet connection is necessary when the system requires access to real-time data from online sources or when it is a mobile-based application. Utilizing a dedicated GPU is ideal for employing deep learning techniques for the system of recommendations, as it may substantially enhance the efficiency of model training. The Python program under the experimental configuration will take in user data and app information, run it through the process for machine learning, and provide app suggestions based on customer tastes and behaviors. The system would furthermore have a feedback loop to enhance suggestions by analyzing user interactions. To confirm the performance of the recommendation system, it will be essential to conduct thorough testing, including both offline testing using historical data and online testing using a live environment.

Figure 6 illustrates a bar graph that compares the average quantities turned based on cap color within a dataset, which may be connected to an iOS or Android recommendation system. In the context of app recommendations, this sort of graph might represent

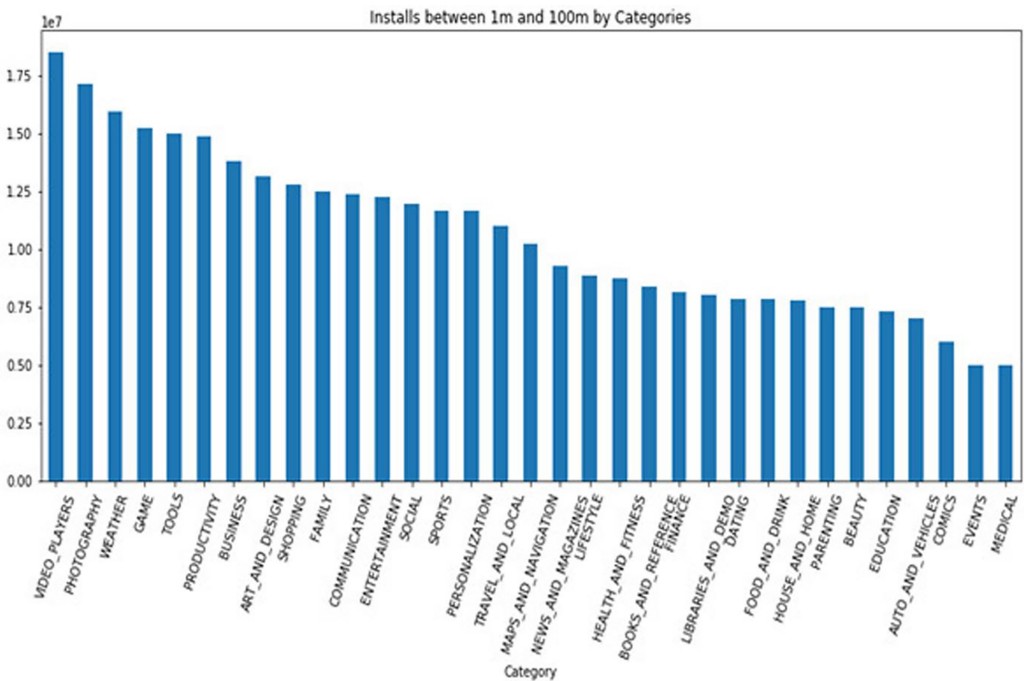

**Figure 6  Categories of mobile app installation.**

multiple categories or features of applications being compared depending on a given statistic, such as user interaction, average ratings, or frequency of usage. In the context of a mobile app recommendation system, the 'cap_color' may symbolically correspond to several app categories (such as gaming, productivity, social networking, *etc.*), while the 'turned' quantity could signify the average user rating for each category. The bars arranged in decreasing order indicate that applications in categories indicated by the leftmost bars, which have larger 'turned' quantities, are often rated higher by users compared to those on the right. Visual data may be utilized in an app system of recommendations to determine the priority of app categories to promote to consumers. Categories with superior ratings or higher levels of interaction may be recommended with more frequency. The visualization enhances the recommendation process by emphasizing user preferences, facilitating the creation of a more customized and efficient app-discovery experience for users. To obtain more accurate interpretations, it is necessary to comprehend the exact measurements and categories depicted in the graph. A graph is a visual representation used to convey trends and patterns in data, which may be used to make informed decisions on in-app recommendations.

## Dataset labeling

A total of 10 k images, both manually and automatically analyzed, make up the dataset. The first thing that automated algorithms did was find images that had logos that didn't match, branding that wasn't aligned, or unique design elements. With these settings, we can identify photographs that are not from the app. Following the automated stage, the highlighted photographs were reviewed by human annotators who made corrections and

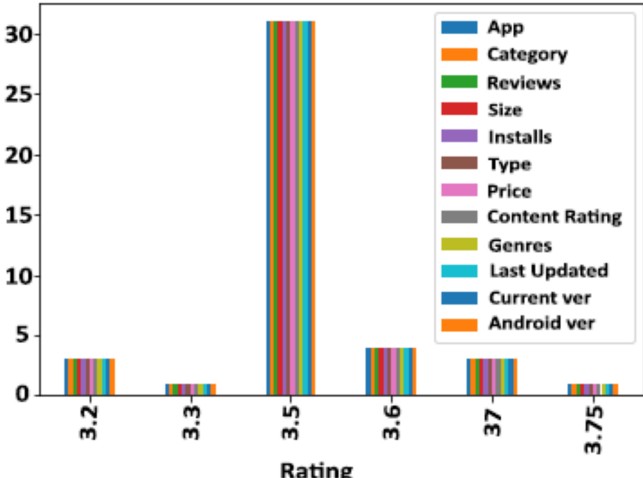

**Figure 7  Data normalization.**    

validated their accuracy. To classify an image as false, we looked for things like the use of illicit icons that looked like popular apps, visual elements that didn't fit the title and criteria of the program, and misleading advertising material. We were able to train the CNN network to detect phony app images by using this tagging technique to generate a valid dataset. Automated systems were the first to notice images with jumbled app logos, mismatched branding, or unusual design components. Before the photos were highlighted, human commentators made sure they were accurate and corrected any false positives. Out of a total of 10,000 data, about 8,000 were authentic and 2,000 unauthentic were not. "Deceptive" means employing visual aspects that don't correspond with the app's purpose or identity, making icons that seem like successful applications, or misleading or erroneous advertising. According to the authors, using a database with confirmed information may make labels more reliable. It may be feasible to detect fake photographs using this database. This method improves the CNN model's ability to recognize fake app visuals by training it on tagged data.

## Implementation and results

This work explains the features, which might include things like strange network behavior, hidden code, overly frequent requests for permission, or anomalous app information or user feedback. We also need to address the issue of whether the apps that were found to be dangerous were indeed detrimental. An illustration of how to create and test apps on an Android OS smartphone using the Android Studio program might be helpful for the Appauthentix algorithm. A thorough validation process and credibility to the findings would be shown by mentioning that certain apps were deemed dangerous depending on the way they behaved during the assessment. We may get a better understanding of the characteristics of malicious programs and the effectiveness of tools for identification by using a comprehensive approach.

Figure 7 displays histograms depicting several data properties derived from a dataset associated with a mobile app ecosystem. These visualizations are commonly employed to

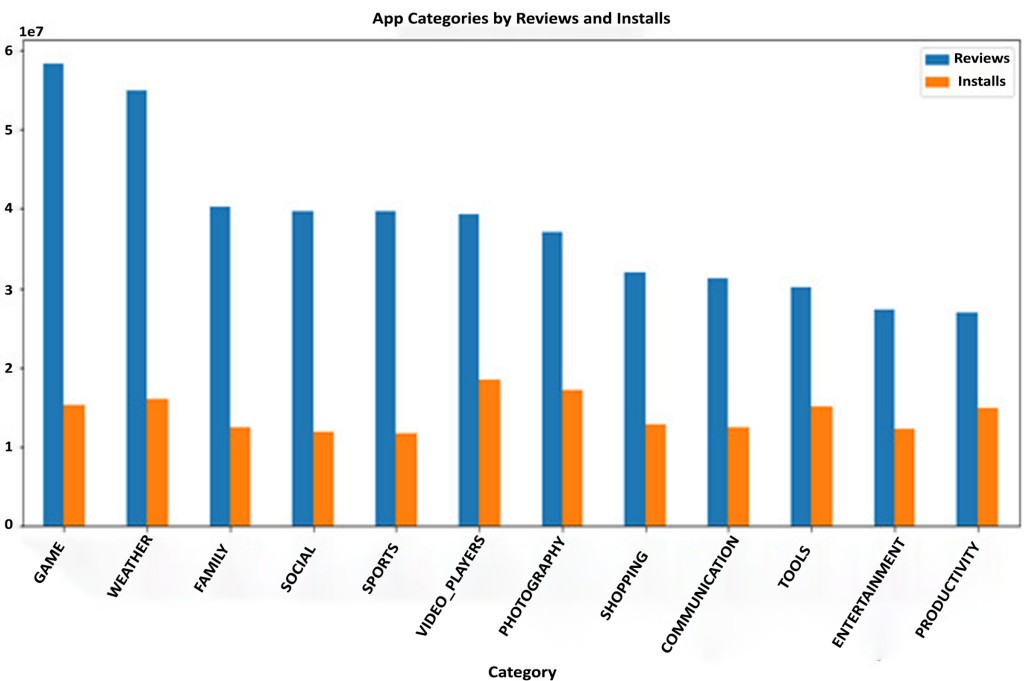

**Figure 8 Categories of mobile app installation and reviews.**

comprehend the distribution, frequency, and interconnections among various variables in a dataset. These visuals might be utilized to comprehend the app market's landscape. For example, if the system detects that a user has a preference for applications in a category that is both highly rated and popular, it might give priority to suggesting additional apps from that same category. Moreover, comprehending the dispersion of installations might facilitate the suggestion of applications that are now popular or have a well-established history of user acceptance. Each graph offers a concise representation of the data, aiding the recommendation engine in generating data-driven choices for consumers. Through the examination of these patterns, the system may customize its suggestions based on the user's specified interests and habits when interacting with the app store.

Figure 8 displays app classifications on the x-axis, ranging from "Game" to "Weather," to indicate the many sorts of applications analyzed. The y-axis presumably represents the count in millions, reflecting the number of reviews and installs for each category, given the average size of app reviews and installs. The graph presents a comparison between two data sets: the number of reviews (depicted in orange) and the number of installations (represented in blue) for each category of applications. The categories "Game" and "Family" have significant figures in terms of both reviews and installations, indicating that both categories are highly sought-after and often downloaded software genres with active user involvement. The categories "Finance" and "Lifestyle" have a reduced quantity of reviews and installations, suggesting that these applications are downloaded less frequently or users are less eager to provide feedback for them. This graph may be utilized in an app recommendation system to determine the most engaging or popular categories among

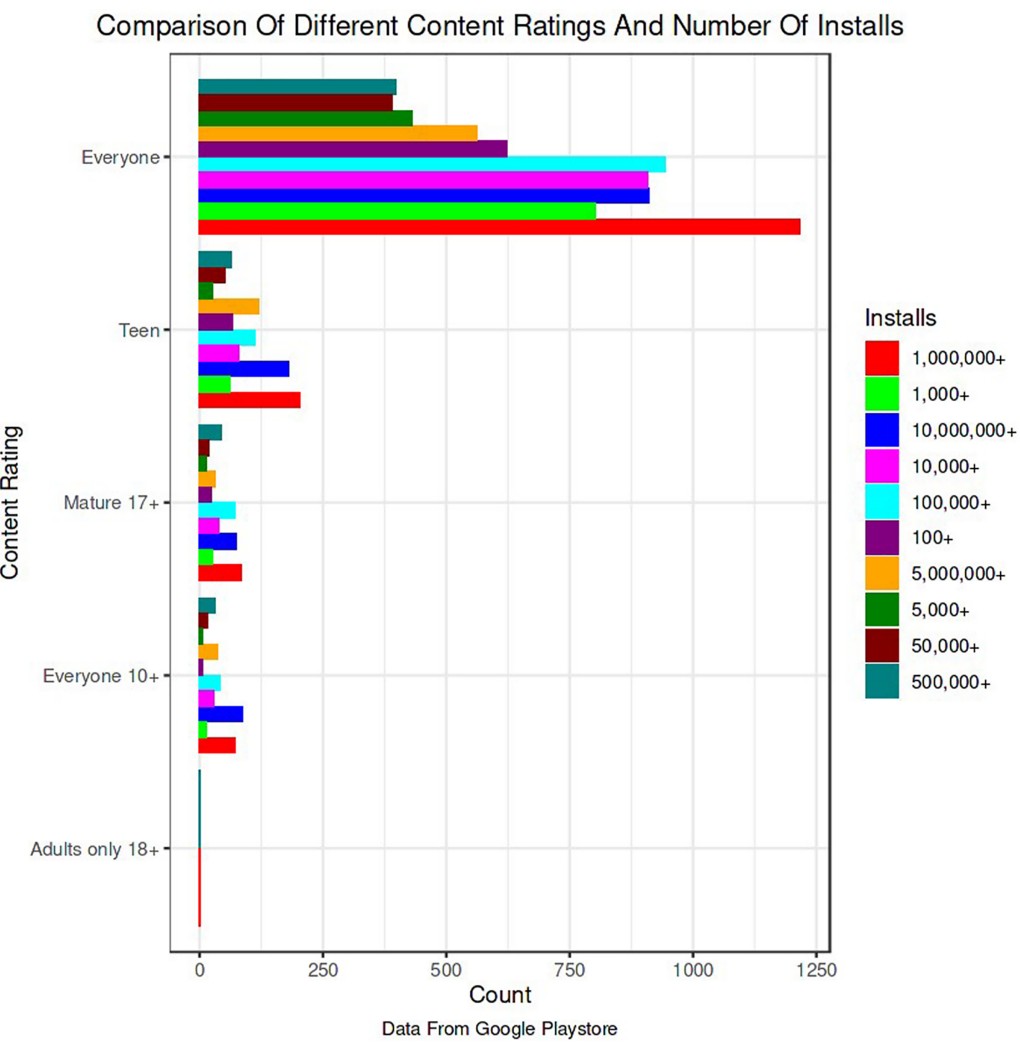

**Figure 9** **Difference between number of install and rating.**

users. This information can then be used to influence the recommendation logic, giving more priority to applications from categories with greater user interaction.

Figure 9 presents a bar chart named "Comparison of Different Content Ratings And Number Of Installs," illustrating the correlation between the content ratings of applications and the number of installations they have. The content ratings, displayed on the y-axis, classify applications based on the suitability of their material for different age groups. These classifications include "Everyone", "Teen", "Mature 17+", and "Adults only 18+". The x-axis represents the "Count" variable, which indicates the number of app installs. Although not specifically labeled, it is likely to be a cumulative count based on the number of applications in each content rating category that fall into various installation brackets. The brackets are color-coded and displayed in the legend on the right side. They range from "1,000,000,000+" installations (in dark blue) to "100+" installations (in red). The bar chart demonstrates that applications categorized as "Everyone" exhibit the most

extensive spectrum of installations, encompassing both the lowest and highest brackets. Furthermore, a substantial proportion of these apps surpass the threshold of 1 billion installations. Applications classified as "Teen" likewise exhibit a wide spectrum of installations, but with a lower number in the highest category. The categories labeled "Mature 17+" and "Adults only 18+" have a somewhat smaller range, with the bulk of games falling within the lower installation brackets. This visualization provides valuable insights into the popularity of different content ratings among users. It may help app developers and marketers determine which content ratings result in the highest number of installations. The most notable observation from the figure is that applications built for the "Everyone" category had the largest reach, which is consistent with the intuitive understanding that more inclusive content ratings enable a wider market penetration. Figure 9 is accompanied by a footer that states "Data From Google Playstore," indicating the origin of the data and providing reassurance to the reader regarding its reliability. Such research might assist in customizing app development and marketing tactics to successfully target the most profitable parts of the app market.

Figure 10 presents a stacked histogram labeled "Number Of App Ratings In Different Top 10 Categories," which visually represents the distribution of app ratings among ten distinct app categories sourced from the Google Play Store. The x-axis corresponds to the rating scale ranging from 1 to 5, including the full spectrum of potential ratings that an app might attain on the Google Play Store. The y-axis displays the number of applications within each rating increase. Every color in the histogram corresponds to a distinct application category, as shown in the legend located on the right side. The categories included are Business, Communication, Family, Game, Lifestyle, Medical, Personalization, Productivity, Sports, and Tools. The colorful stacks visually represent the number of apps in each category that belong to each rating range. The graph indicates that a significant proportion of applications in these categories receive high ratings, with the highest concentration of ratings falling between 4 and 4.5. The category labeled 'Game' stands out due to its abundance of top-rated applications, as seen by the intensity of its color in the rating range of 4 to 5. In contrast, the graph demonstrates a decrease in the number of applications with poor ratings, as indicated by the shorter bars on the left side of the graph, namely in the 1–2 rating range. The histogram is a graphical representation that succinctly presents the relative popularity and user satisfaction of applications in different categories. This information may be quite beneficial for app developers and marketers as it allows them to comprehend market trends and pinpoint areas that require enhancement. The presence of the notation "Data From Google Playstore" at the bottom serves to emphasize the origin of the data, assuring the reader that the information is derived directly from the marketplace where these programs are distributed.

Figure 11 displays a correlation matrix heatmap, which visually represents the correlation coefficients between pairs of variables in a dataset. This is our metric for evaluating classification accuracy. The heatmap displays the variables 'Rating', 'Reviews', 'Size', 'Installs', and 'Price' along both the x-axis and the y-axis. Every individual cell inside the grid reflects the correlation between two variables. The level of correlation is shown by both the intensity of color and the numerical value displayed within the cell. The

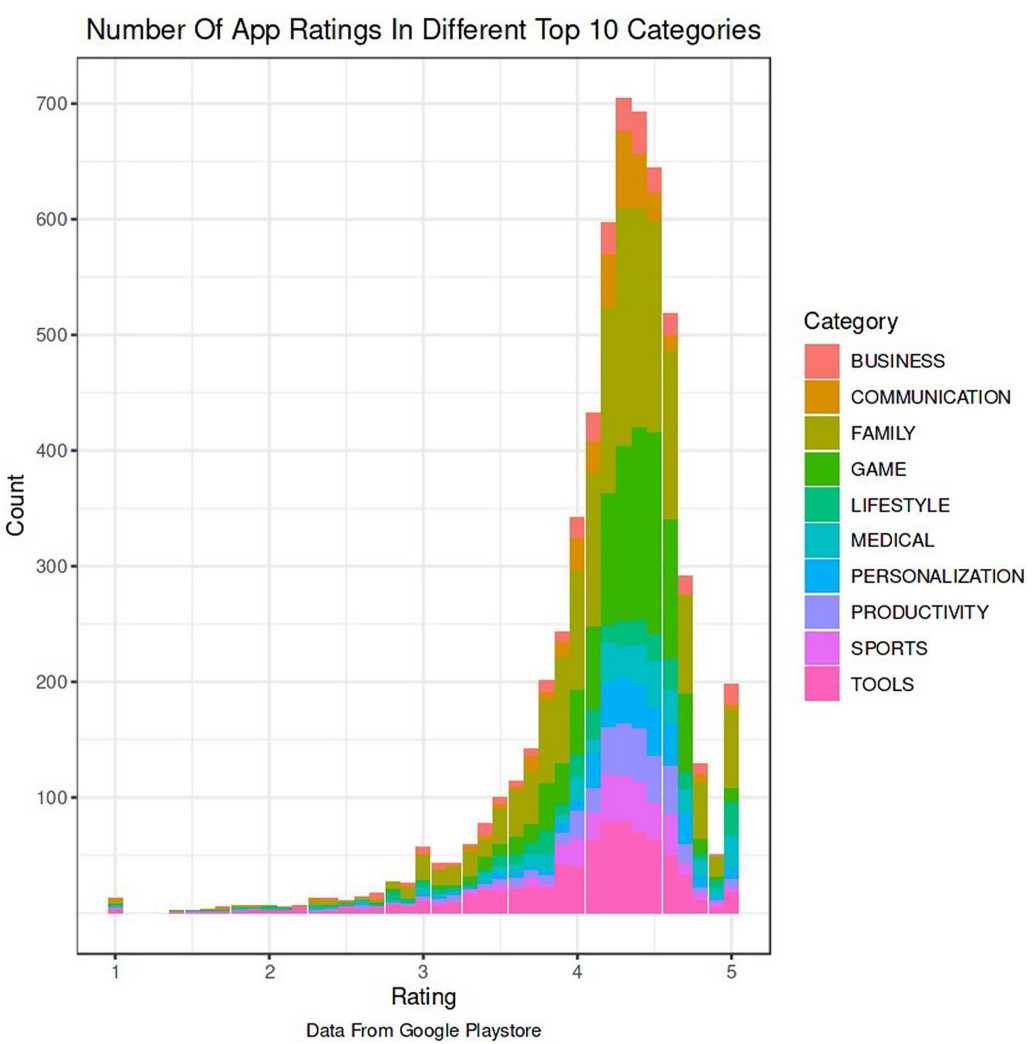

Number Of App Ratings In Different Top 10 Categories

**Figure 10  App rating classification.**

correlation values span from −1 to 1, with 1 being a flawless positive correlation, −1 representing a flawless negative correlation, and 0 representing no connection. In this specific heatmap, deeper hues of one hue (often purple or blue) signify more pronounced positive connections, whereas deeper hues of another hue (typically red or orange) imply more pronounced negative correlations. For example, when a cell intersects the variables 'Rating' and 'Reviews' and has a value of 0.1, it indicates a little positive correlation. This means that applications with higher ratings generally tend to have somewhat more reviews. Conversely, when the 'Reviews' and 'downloads' variables overlap with a value of 0.6, it indicates a moderate positive correlation. This implies that programs with a higher number of reviews generally have a higher number of downloads. The diagonal line of cells from the top left to the bottom right exhibits perfect correlations of 1.0 since it represents the intersection of each variable with itself. A heatmap of this kind is valuable for promptly comprehending the associations between distinct variables and can aid in data analysis to discern patterns or to formulate conclusions predicated on such connections. For instance,

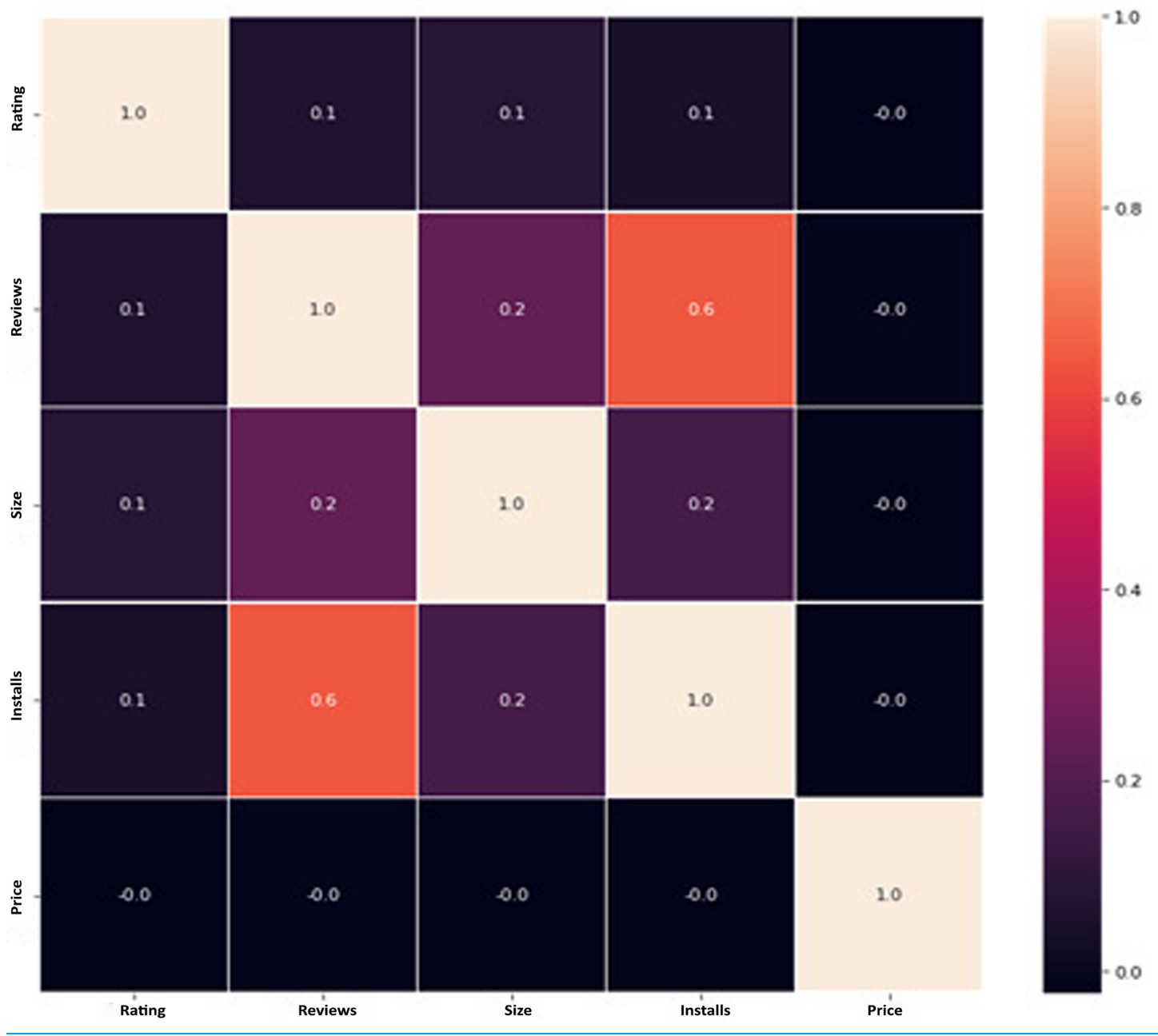

**Figure 11 Correlation matrix.**

if there was a robust positive connection between the variables 'Price' and 'Rating', it might be deduced that customers tend to consider applications with higher prices as being of superior quality. Nevertheless, the heatmap indicates that there is minimal or negligible connection between the 'Price' and 'Rating' variables, implying that the price does not significantly influence an app's rating.

Figure 12 presents a comparative analysis of three distinct technologies or models, namely CNN, NLP, and AppAuthentix Recommender. The comparison is based on three

*Peer*J Computer Science

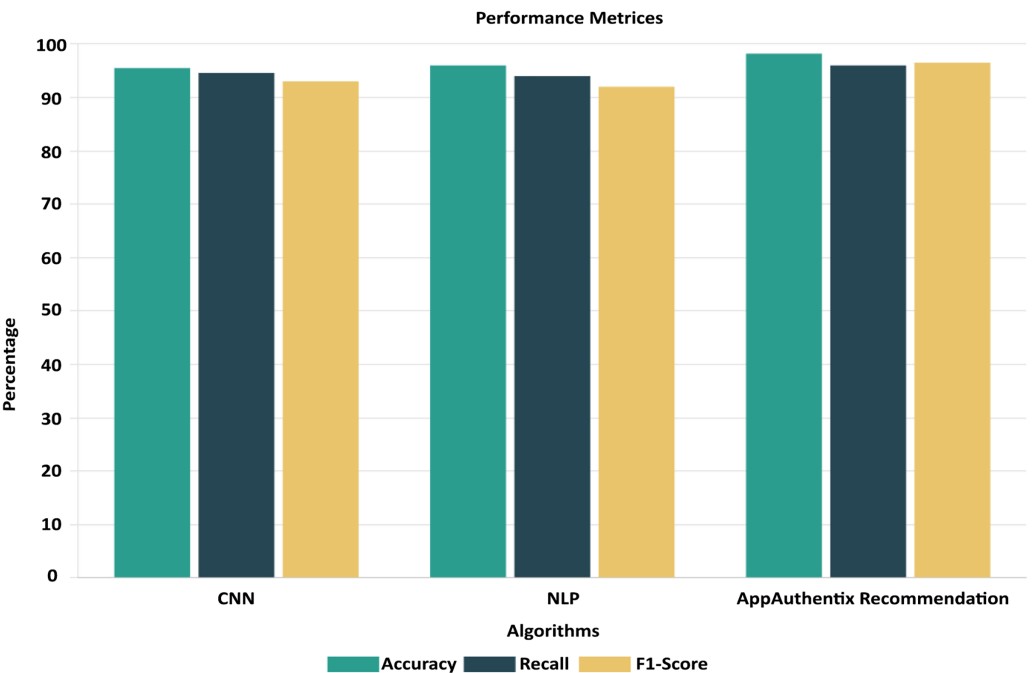

**Figure 12 Difference between existing and proposed algorithm comparison.**

performance metrics: accuracy, recall, and F1-score. Accuracy is the ratio of correct forecasts to total predictions. Recall, also known as sensitivity, quantifies the ratio of accurately detected real positives. The F1-score is a statistical measure that calculates the harmonic mean of accuracy and recall. It aims to strike a balance between these two metrics. Within the graph, each model is shown by three distinct bars, each of which is colored differently to symbolize a certain measure. The AppAuthentix Recommender model has superior accuracy and recall compared to the other two models, suggesting its exceptional ability to accurately identify positive situations and make the right recommendations overall. The F1-score of this model is similar to that of the other models, indicating a favorable equilibrium between accuracy and recall. Although the AppAuthentix Recommender achieves the greatest level of accuracy, it does not exhibit a substantial advantage over CNN or NLP in terms of F1-score. This suggests that while it is accurate, it does not consistently achieve a balance between accuracy and recall as well as the other models. A graph of this nature will provide stakeholders with useful insights to ascertain the optimal model based on the criteria that are most pertinent to their requirements. For example, if the consequences of a false negative are expensive, they may give priority to a model with the highest recall rate. If the objective is to assess the overall performance, one may consider evaluating the F1-Score or a composite of other measures. Figure 13 displays the ROC-AUC values, which were obtained from a comparative study of many machine learning models conducted using Dataiku.
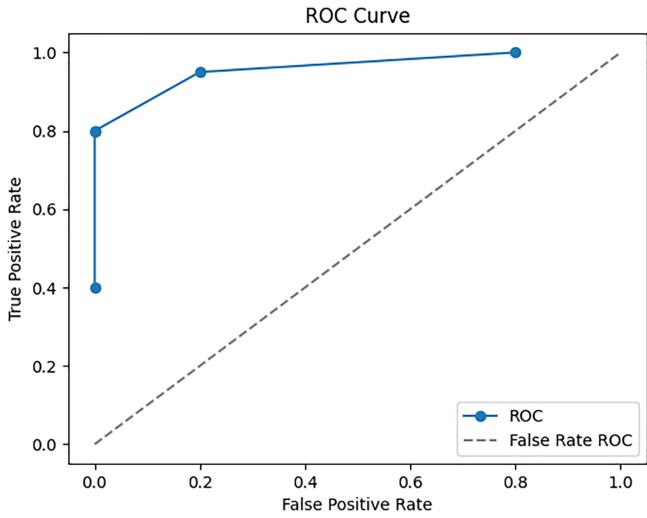

**Figure 13 ROC curve.**               

## Google play's current fraud detection methods with the existing fraud detection

- **Detection techniques:** Analyze the algorithms and technologies used by Google Play, such as machine learning models, behavior analysis, and heuristic methods, and compare them to those employed by other systems. Highlight any unique approaches that either system uses.

- **Accuracy and efficiency:** Examine metrics like false positive rates, detection speed, and overall effectiveness in identifying fraudulent activity. If available, present data or case studies that illustrate performance differences.

- **Adaptability:** Discuss how each system adapts to evolving fraud tactics. Google Play may have specific updates or features that allow it to respond quickly to new threats, while other systems may rely on more static models.

- **User experience:** Consider how the user experience differs between the systems. For example, Google Play's methods should balance security with minimal disruption to legitimate users, while other systems may prioritize security at the cost of user convenience.

- **Integration and collaboration:** Evaluate how well each system integrates with other security measures or platforms. Google Play might leverage its vast ecosystem for data sharing, while other systems may operate more independently.

### Performance metrics

Accuracy, precision, recall, and the F1 score were all used to evaluate the effectiveness of our proposed method. The four measures used to calculate these metrics are true positive (TP), true negative (TN), false positive (FP), and false negative (FN).

   **TP:** The ratio of samples properly identified by the method of detection model to the total number of samples.

**TN:** The fraction of samples for which the detection model's classification of their true type is accurate.

**FP:** The actual sample type is normal; however, the detection model incorrectly identified a large number of samples as coming from a DDoS attack.

**FN:** The amount of DDoS attack samples that were incorrectly classified as "normal" samples.

**Accuracy:** This represents the proportion of input samples for which the detection model reached a positive verdict as shown in Eq. (12).

$$AC = \frac{TP + TN}{TP + TN + FP + FN} \qquad (12)$$

**Recall:** It is the proportion of DDoS attack samples accurately identified by an identification model out of the total number of attack samples as shown in Eq. (13).

$$R = \frac{TP}{TP + FN} \qquad (13)$$

**Precision:** It represents the proportion of samples that the detection model has identified as being subject to a DDoS attack as shown in Eq. (14).

$$P = \frac{TP}{TP + FP} \qquad (14)$$

**F1-score:** It is an aggregate measure of accuracy that takes into account both recall and precision as shown in Eq. (15).

$$F_1 = \frac{2}{\frac{1}{P} + \frac{1}{R}} \qquad (15)$$

## CONCLUSION

To summarize, this work is a significant advancement in tackling the crucial issue of detecting and fighting counterfeit and harmful apps in the fast-growing mobile application industry. The incorporation of advanced technologies such as CNN, NLP, and the innovative AppAuthentix Recommender algorithm has resulted in a revolutionary approach that enhances app store security and boosts user trust in the digital marketplace. The increasing spread of counterfeit and detrimental software has presented significant hazards to both consumers and the credibility of app marketplaces. The research has addressed this urgent requirement by creating a sophisticated predictive model that integrates image analysis with text-based feature extraction. This model achieves an amazing identification accuracy rate of 98.25% in differentiating authentic mobile applications from counterfeit ones. The rationale for this study arises from the paramount significance of ensuring the security of app marketplaces and protecting consumers from deceitful and detrimental programs. Through the utilization of these cutting-edge technologies, our goal is to create a more secure digital atmosphere for consumers of mobile applications, while simultaneously fostering confidence in app marketplaces. The

outcomes of our investigation are quite extraordinary, showcasing a level of precision in predicting accuracy that establishes a groundbreaking benchmark in app store security. This accomplishment not only strengthens the safeguarding of users but also establishes the foundation for future progress in mobile app authentication. To summarize, this research provides an innovative and thorough answer to the widespread problem of identifying apps during a period of extraordinary expansion in mobile applications. The use of CNN, NLP, and the AppAuthentix Recommender algorithm represents a significant advancement in improving the security of app stores. By adopting these cutting-edge methods, we can guarantee a better-protected digital environment and strengthen user confidence in mobile app platforms, ultimately cultivating a safer and more reliable digital terrain for everyone.

## FUTURE WORK

In future endeavors, the use of federated learning methods has the potential to significantly improve the security and precision of mobile app identification within the aforementioned environment. Federated learning is a distributed machine learning method that enables collaborative training of models across many devices or platforms while ensuring the protection of user privacy. By integrating federated learning into the app identification architecture, several breakthroughs may be attained. Firstly, it allows for immediate upgrades and enhancements to be implemented across different app stores and user devices, ensuring that the identification model stays flexible and responsive to new and emerging threats. In addition, federated learning can improve user privacy by ensuring that sensitive data is stored on individual devices, thereby mitigating privacy concerns related to centralized data collecting. In addition, federated learning enables the transmission and application of insights and patterns from one app store to another, enhancing the app identification system by making it more robust and complete. This methodology can greatly enhance the precision of detecting both genuine and counterfeit applications, while continuously adjusting to emerging methods of attack. Further research on federated learning methods has the potential to enhance the mobile app identification system, making it more adaptable, privacy-oriented, and precise. This would contribute to creating a digital environment that is safer and more reliable for consumers.

## ACKNOWLEDGEMENTS

I need to convey my sincere thanks to Deep Learning Laboratory, Department of Artificial Intelligence and Data Science, Ramco Institute of Technology.

### Funding

The authors received no funding for this work.

### Competing Interests

The authors declare that they have no competing interests.

## Author Contributions

- Ramnath M. conceived and designed the experiments, performed the experiments, analyzed the data, performed the computation work, prepared figures and/or tables, authored or reviewed drafts of the article, and approved the final draft.
- Yesubai Rubavathi C. analyzed the data, authored or reviewed drafts of the article, and approved the final draft.

## Data Availability

The Analysis of App Recommendations and User Reviews dataset is available in the Supplemental File and at Mendeley Data: RAMNATH (2024), "Analysis of App Recommendations and User Reviews", Mendeley Data, V1, doi: 10.17632/2p8w7p8kty.1.

The Mobile Applications images & Logos Dataset is available in the Supplemental File and at Mendeley Data: M, RAMNATH (2024), "Mobile Applications images & Logos", Mendeley Data, V1, doi: 10.17632/nvxjm84n6f.1.

## Supplemental Information

Supplemental information for this article can be found online at http://dx.doi.org/10.7717/peerj-cs.2515#supplemental-information.

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
