# Peer review of "Enhancing AppAuthentix recommender systems using advanced machine learning techniques to identify genuine and counterfeit android applications"

_PeerJ Computer Science, doi:10.7717/peerj-cs.2515_

## Round 0.1 · original submission · Major Revisions

Please address all reviewer comments

Reviewer 1 ·

Basic reporting

- The paper introduces a novel solution integrating Convolutional Neural Networks (CNN), Natural Language Processing (NLP), and the AppAuthentix Recommender algorithm, showcasing a sophisticated and comprehensive method for app identification and security.
- The integration of CNN and NLP for app identification is a novel approach. The study addresses a critical issue in the app ecosystem, providing a potentially impactful solution.
- The authors are suggested to enhance the grammar and sentence structure throughout the article to improve readability and comprehension.
- The full forms of abbreviations in the abstract have not been provided; the full forms of the abbreviations can be given in the abstract. Ensure consistency in the usage of abbreviations and their full forms throughout the article.
- The quality of the figures need to be improved. Authors are suggested to consider enhancing the quality of the figures to ensure they are more visually effective and informative.
- A subsection can be added to discuss security aspects of the proposed solution.
- While the paper references several related works, it could benefit from a more in-depth and critical review of existing literature. To this end, a table can be added to compare related works and the proposed solution to better highlight the contributions and novelty of the work.

Experimental design

- A more detailed description of the CNN and NLP models used, including their architectures, hyperparameters, and training procedures, need to be provided.
- Explain the data preprocessing steps in greater detail, especially how text and image data are prepared for analysis.
- A section on the validation and testing procedures, specifying how the model performance was evaluated, can be added.

Validity of the findings

-A discussion on the potential limitations of the datasets and how they might affect the results can be added.
- Empirical data on user satisfaction and trust to support the claims made about the impact of the proposed system can be included
- A discussion can be added for ethical and privacy aspects of the proposed solution.
- A more extensive discussion on the evaluation metrics and their limitations can be added to provide a clearer understanding of the proposed solution’s robustness.
- The practical implications and real-world applications of the proposed solution can be added to enhance the relevance and impact of the research.

·

Basic reporting

The paper, as I understand it is about detecting malicious or fraudulent apps. The authors do not have a canonical definition of malicious behavior. They do not define the characteristics of malicious apps they are interested in detecting. Hence, the term fraudulent is ambiguous here.

Moreover, the paper has a lengthy discussion on orthogonal topics such as accessibility. From my understanding of the paper's focus, I believe such discussions are not necessary as they distract the reader from the core issue being addresses, i.e., detecting malicious/fraudulent apps.

Finally, the paper lacks a discussion on other approaches used to detect malicious apps such as static and dynamic analysis tools. They are computationally cheaper than NLP and CNN-based tools. It will be useful for the reader to understand how the proposed tool compares with state-of-the static/dynamic analysis tools, many of which are already used by app stores to detect malicious apps.

Experimental design

The paper lacks an explanation of how the CNN dataset was labelled. Were the images of all 10k apps manually annotated as authentic and fraudulent? What factors led to classifying an image as fraudulent? The authors should clearly delineate the method of labeling the dataset along with the number of apps labeled as authentic and fraudulent. They should also clearly define what they mean by fraudulent. I think using an established repository of ground truths might help here.

Validity of the findings

The comparison with NLP and CNN is good. However, it is difficult to understand from the current version of the paper what features of an app make it malicious. The authors should consider adding a discussion on the features that make an app potentially malicious. Also, there is no discussion on whether the apps detected are truly malicious. Did the authors verify their malicious behavior by running them on a device or emulator. If yes, they should present that information in the paper.

Additional comments

Minor comments:

- “The efficacy of CNN in identifying counterfeit applications may largely be assessed inside constrained situations or specialized app markets.” Authors haven’t cited anything to support this statement.

- Authors don’t explain why they chose Kaggle dataset instead of datasets like AndroZoo. What was the rationale behind choosing a dataset should be explained.

·

Basic reporting

In this paper, the authors propose an advanced machine learning model to distinguish genuine Android applications from counterfeit ones. They achieve this by integrating both picture analysis and text-based feature extraction. My main concern revolves mostly around the structure of the paper. For instance, it is somewhat unconventional to use "amazing results" in the abstract. Additionally, why does the introduction section resemble a literature review? This section should instead introduce the field, clearly articulate the problem, and outline the authors' contributions using bullet points such as "to evaluate," "to develop," "to implement," etc. It is also important to discuss the structure of the paper at the end of the introduction section. Reviewing existing published papers will provide guidance on how to proceed with this. Additionally, the gap in the literature review is rather unclear, the reflection on existing work should be critical to justify your approach.

Experimental design

The experimental results are scattered. Why did you not include figures in this section and move them to after the references section and appendix? Please place them immediately after discussing each figure, as otherwise, the reader will need to jump between the results and appendix to match the figures with what you claimed in your proposal. This section is also short; I believe materials from the previous section should be included here.

Validity of the findings

The results are promising and the approach is novel. However, the justification is rather unclear, and the entire document, including the findings, should be restructured.

Additional comments

The structure of the paper is the issue. It should be changed as it has a rather unusual structure. Perhaps follow this format: Abstract, Introduction, Related Work, Methodology, Design & Analysis, Implementation & Results, Conclusion & Future Work, References.

---

## Round 0.2 · Minor Revisions

Please revise the article accordingly and resubmit. nr

Reviewer 1 ·

Basic reporting

-The authors addressed my previous comments successfully.
- While the full forms of abbreviations are now provided in the abstract, ensure this consistency is maintained throughout the text, especially in the introduction and related work sections.
-The authors are advised to increase the font size of Figures 2, 3, 7, 8, 11, and 12 to improve their clarity. In particular, Figure 3 could benefit from being redesigned to enhance its visual presentation and comprehensibility.

Experimental design

- The authors have provided a more detailed description of the CNN and NLP models, but the explanation of the hyperparameters remains somewhat superficial. Including the rationale behind the choice of hyperparameters (such as learning rate, batch size, etc.) would provide a clearer understanding of the experimental setup.
-The validation and testing procedures are now clearer. However, a more explicit discussion on the choice of evaluation metrics (accuracy, F1-score, etc.) and why they were selected for this task would further strengthen the paper.

Validity of the findings

-The practical implications of this system are clearer in the revised manuscript. A brief discussion on how this system could be integrated into existing app store ecosystems like Google Play or how it compares to current fraud detection systems could further strengthen the relevance of the research.

---

## Round 0.3 · accepted · Accept

The article is now at a very good level and can be accepted for publication